# An H3K9 methylation-dependent protein interaction regulates the non-enzymatic functions of a putative histone demethylase

Gulzhan Raiymbek[1], Sojin An[1], Nidhi Khurana[1], Saarang Gopinath[1], Ajay Larkin[1], Saikat Biswas[1], Raymond C Trievel[1,2], Uhn-soo Cho[1,2], Kaushik Ragunathan[1]*

[1]Department of Biological Chemistry, University of Michigan, Ann Arbor, United States; [2]Department of Biophysics, University of Michigan, Ann Arbor, United States

**Abstract** H3K9 methylation (H3K9me) specifies the establishment and maintenance of transcriptionally silent epigenetic states or heterochromatin. The enzymatic erasure of histone modifications is widely assumed to be the primary mechanism that reverses epigenetic silencing. Here, we reveal an inversion of this paradigm where a putative histone demethylase Epe1 in fission yeast, has a non-enzymatic function that opposes heterochromatin assembly. Mutations within the putative catalytic JmjC domain of Epe1 disrupt its interaction with Swi6[HP1] suggesting that this domain might have other functions besides enzymatic activity. The C-terminus of Epe1 directly interacts with Swi6[HP1], and H3K9 methylation stimulates this protein-protein interaction in vitro and in vivo. Expressing the Epe1 C-terminus is sufficient to disrupt heterochromatin by outcompeting the histone deacetylase, Clr3 from sites of heterochromatin formation. Our results underscore how histone modifying proteins that resemble enzymes have non-catalytic functions that regulate the assembly of epigenetic complexes in cells.

*For correspondence:
ragunath@umich.edu

Competing interests: The authors declare that no competing interests exist.

## Introduction

The covalent and reversible modification of histones allows cells to establish stable and heritable patterns of gene expression without any changes to their genetic blueprint. Histone H3 lysine nine methylation (H3K9me) is a conserved mark of transcriptional silencing that is associated with the formation of specialized domains called heterochromatin. Heterochromatin formation is critical for centromere and telomere function, silencing of transposons and repetitive DNA elements, and for maintaining lineage-specific patterns of gene expression (*Grewal and Jia, 2007*; *Nicetto et al., 2019*). It is widely assumed that a delicate interplay between histone modification readers, writers, and erasers regulates the establishment and maintenance of epigenetic states (*Allis and Jenuwein, 2016*). In fission yeast, the establishment of H3K9 methylation requires the enzymatic activity of a conserved methyltransferase, Clr4[Suv39h] (*Rea et al., 2000*). H3K9 methylation acts as a platform to recruit chromatin effector proteins with reader domains. Two HP1 homologs, Swi6[HP1] and Chp2[HP1], are involved in H3K9 methylation binding and have distinct, non-overlapping functions during transcriptional silencing (*Motamedi et al., 2008*). Ultimately, factors such as Epe1, a putative histone demethylase, promote epigenetic erasure, thus reversing H3K9 methylation and transcriptional gene silencing (*Ayoub et al., 2003*; *Trewick et al., 2005*).

The sequence-specific recruitment of Clr4[Suv39h] restricts heterochromatin establishment to distinct sites in the genome, such as the centromeres, telomeres, and the mating-type locus (*Bayne et al., 2010*; *Hall et al., 2002*; *Verdel et al., 2004*; *Volpe et al., 2002*). Once established,

**eLife digest** A cell's identity depends on which of its genes are active. One way for cells to control this process is to change how accessible their genes are to the molecular machinery that switches them on and off. Special proteins called histones determine how accessible genes are by altering how loosely or tightly DNA is packed together.

Histones can be modified by enzymes, which are proteins that add or remove specific chemical 'tags'. These tags regulate how accessible genes are and provide cells with a memory of gene activity. For example, a protein found in yeast called Epe1 helps reactivate large groups of genes after cell division, effectively 're-setting' the yeast's genome and eliminating past memories of the genes being inactive.

For a long time, Epe1 was thought to do this by removing methyl groups, a 'tag' that indicates a gene is inactive, from histones – that is, by acting like an enzyme. However, no direct evidence to support this hypothesis has been found. Raiymbek et al. therefore set out to determine exactly how Epe1 worked, and whether or not it did indeed behave like an enzyme.

Initial experiments testing mutant versions of Epe1 in yeast cells showed that the changes expected to stop Epe1 from removing methyl groups instead prevented the protein from 'homing' to the sections of DNA it normally activates. Detailed microscope imaging, using live yeast cells engineered to produce proteins with fluorescent markers, revealed that this inability to 'home' was due to a loss of interaction with Epe1's main partner, a protein called Swi6. This protein recognizes and binds histones that have methyl tags. Swi6 also acts as a docking site for proteins involved in deactivating genes in close proximity to these histones.

Further biochemical studies revealed how the interaction between Epe1 and Swi6 can help in gene reactivation. The methyl tag on histones in inactive regions of the genome inadvertently helps Epe1 interact more efficiently with Swi6. Then, Epe1 can simply block every other protein that binds to Swi6 from participating in gene deactivation. This observation contrasts with the prevailing view where the active removal of methyl tags by proteins such as Epe1 switches genes from an inactive to an active state.

This work shows for the first time that Epe1 influences the state of the genome through a process that does not involve enzyme activity. In other words, although the protein may 'moonlight' as an enzyme, its main job uses a completely different mechanism. More broadly, these results increase the understanding of the many different ways that gene activity, and ultimately cell identity, can be controlled.

H3K9 methylation spreads to silence genes that are distant from heterochromatin nucleation centers. H3K9me spreading depends on the read-write activity of Clr4[Suv39h] (*Al-Sady et al., 2013*; *Zhang et al., 2008*). Two conserved, structural attributes of Clr4[Suv39h] mediate this process: a conserved chromodomain that recognizes and binds to H3K9me and an enzymatic SET domain that is involved in catalysis (*Ivanova et al., 1998*). The ability of Clr4[Suv39h] to bind to the product of its enzymatic activity enables H3K9 methylated histones to serve as carriers of epigenetic information (*Audergon et al., 2015*; *Ragunathan et al., 2015*). Following DNA replication, H3K9 methylated histones that are partitioned between daughter DNA strands serve as templates to mark newly deposited histones.

H3K9 methylation acts as a multivalent platform that recruits the HP1 homolog, Swi6[HP1] to sites of constitutive heterochromatin (*Ekwall et al., 1995*; *Ekwall et al., 1996*). HP1 proteins have a conserved architecture consisting of a chromodomain (CD) that is involved in H3K9 methylation binding and a dimerization domain called the chromoshadow domain (CSD) that mediates protein-protein interactions (*Bannister et al., 2001*; *Cowieson et al., 2000*). The oligomerization of HP1 proteins promotes formation of higher-order chromatin complexes that exhibit phase separated, liquid-like properties in vitro and in cells (*Larson et al., 2017*; *Sanulli et al., 2018*; *Strom et al., 2017*). Swi6[HP1] interacts with a broad spectrum of agonists and antagonists that influence epigenetic silencing. Most notably, recruitment of Epe1, a putative histone demethylase that opposes heterochromatin formation, is dependent on Swi6[HP1] (*Ayoub et al., 2003*; *Zofall and Grewal, 2006*). Loss of Epe1 leads to increased heterochromatin spreading beyond normal boundary sequences, inheritance of

H3K9 methylation via sequence-independent pathways, increased cell to cell variation in H3K9 methylation patterns, and acquisition of adaptive epigenetic traits (*Audergon et al., 2015*; *Ayoub et al., 2003*; *Ragunathan et al., 2015*; *Sorida et al., 2019*; *Trewick et al., 2007*; *Wang et al., 2015*; *Zofall and Grewal, 2006*; *Zofall et al., 2012*). Despite its critical role in heterochromatin regulation, how Epe1 exerts its anti-silencing function in cells remains mysterious (*Trewick et al., 2007*).

Histone demethylases have prominent roles in regulating the reversibility of epigenetic states (*Iwase et al., 2007*; *Whetstine et al., 2006*). Changes in their expression lead to widespread chromatin reorganization, which alters both the prognosis and treatment of diseases such as cancer (*Liau et al., 2017*). Epe1 most closely resembles JumonjiC (JmjC) domain containing proteins that use Fe (II) and α-ketoglutarate as co-factors to catalyze histone demethylation (*Tsukada et al., 2006*). Unlike active histone demethylases, where an HxD/E….H motif is involved in Fe (II) coordination, Epe1 harbors a non-canonical HXE….Y motif (*Tsukada et al., 2006*). Although Epe1 shares conserved features with other histone demethylases at the amino acid level, there is no biochemical evidence to support the notion that Epe1 has any in vitro enzymatic activity. Epe1 purified from fission yeast cells, or a recombinant source (insect cells), exhibits no H3K9 demethylase activity in vitro (*Tsukada et al., 2006*; *Zofall and Grewal, 2006*). However, point mutations of amino acid residues involved in Fe (II) or α-ketoglutarate binding affect Epe1 activity in cells and lead to heterochromatin spreading beyond normal boundary sequences (*Trewick et al., 2007*). These conflicting lines of biochemical and genetic data have prompted several alternative explanations for how Epe1 might fulfill its anti-silencing role. These models include the possibility that Epe1 acts as a protein hydroxylase which targets non-histone proteins such as Swi6$^{HP1}$, regulates the activity of the multi-subunit H3K9 methyltransferase CLRC complex, or functions as an H3K9 demethylase when in complex with Swi6$^{HP1}$ (*Aygün et al., 2013*; *Iglesias et al., 2018*; *Trewick et al., 2007*; *Zofall and Grewal, 2006*). Although these models represent attractive possibilities for how Epe1 regulates heterochromatin, there is no direct evidence to suggest that any of these proteins represent bonafide enzymatic targets. An alternative hypothesis is that Epe1 has a non-enzymatic function that regulates heterochromatin spreading and epigenetic inheritance. In support of this hypothesis, the overexpression of Fe (II) and α-ketoglutarate binding mutants of Epe1 suppresses heterochromatin spreading defects observed in *epe1Δ* strains (*Trewick et al., 2007*; *Zofall and Grewal, 2006*). Hence, co-factor binding mutants of Epe1 can act as multi-copy suppressors of epigenetic silencing despite the presumptive loss of enzymatic activity.

In this study, we discovered that the putative catalytic JmjC domain of Epe1 is, at least in part, dispensable for its anti-silencing function in cells. The C-terminus of Epe1 directly interacts with Swi6$^{HP1}$ and its interaction in the context of full-length Epe1 is regulated by H3K9 methylation. Expressing the Epe1 C-terminus alone is sufficient to reverse heterochromatin establishment and attenuate epigenetic inheritance. We propose that a *cis* interaction between the Epe1 N- and C-terminus inhibits Swi6$^{HP1}$ binding. H3K9 methylation binding attenuates this intramolecular interaction and promotes Swi6$^{HP1}$ binding. A requirement for H3K9 methylation to stabilize a complex comprising Epe1 and Swi6$^{HP1}$ restricts their interaction to a heterochromatin-specific context. Our work highlights the versatile, non-canonical ways in which histone demethylases can oppose establishment and maintenance of epigenetic states.

## Results

### A point mutation within the catalytic JmjC domain of Epe1 affects its localization at sites of constitutive heterochromatin

JmjC domain-containing proteins require Fe (II) and α-ketoglutarate as co-factors to catalyze histone demethylation. Aligning the primary amino acid sequences of active histone demethylases with Epe1 reveals a naturally occurring histidine to tyrosine substitution (Y370) within a conserved triad of amino acid residues that coordinate iron (*Figure 1—figure supplement 1A*). We tested whether the activity of Epe1 in cells is dependent on this non-conserved tyrosine residue (Y370). To measure Epe1 activity, we used a reporter gene assay that provides a direct read-out of epigenetic inheritance. In this system, an H3K9 methyltransferase, Clr4$^{Suv39h}$ is fused to a DNA binding protein, TetR. This fusion protein is recruited to an ectopic site where ten Tet operator sites (*10X TetO*) are placed upstream of a reporter gene, *ade6+* (*Figure 1A*). Establishment in the absence of tetracycline results

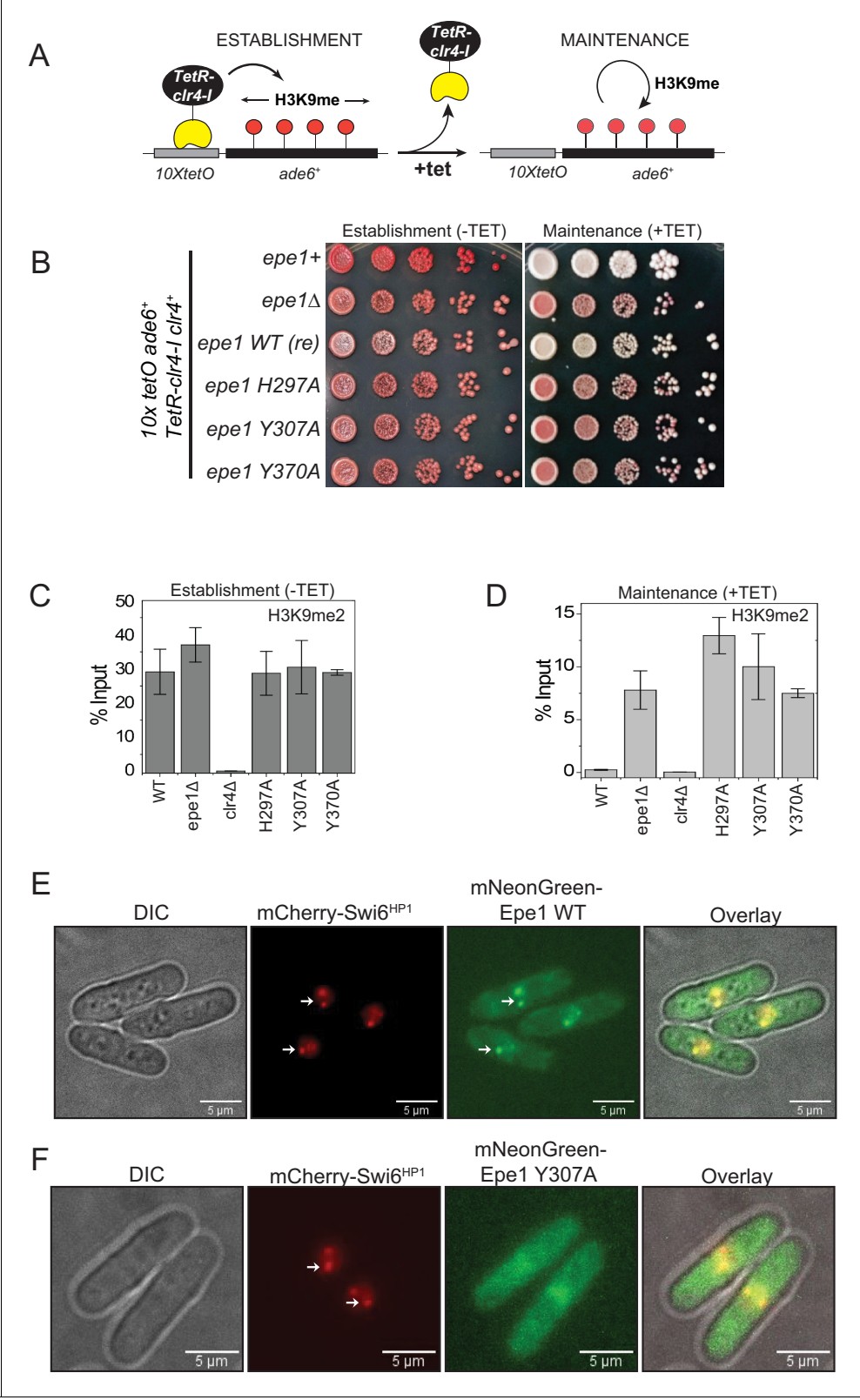

**Figure 1.** Point mutation within the catalytic JmjC domain of Epe1 affects protein localization at sites of constitutive heterochromatin. (**A**) Reporter system to measure epigenetic inheritance. TetR-Clr4-I binding (−tetracycline) leads to ectopic establishment of H3K9 methylation. Addition of tetracycline promotes TetR-Clr4-I dissociation, enabling us to measure epigenetic inheritance of H3K9 methylation. (**B**) Color-based assay to detect establishment and maintenance of epigenetic states. The establishment of epigenetic silencing (−tetracycline) leads to the appearance of red colonies. *Figure 1 continued on next page*

*Figure 1 continued*

Epigenetic inheritance, indicated by red or sectored colonies, (+tetracycline) is critically dependent on Epe1 activity. Point mutations within the JmjC domain of Epe1 disrupt its anti-silencing function in cells, leading to the appearance of red or sectored colonies. (C) ChIP-qPCR measurements of H3K9me2 levels at the ectopic site (*10X tetO ade6+*) during establishment (−tetracycline) in different Epe1 mutant backgrounds (N = 2). Error bars represent standard deviations. (D) ChIP-qPCR measurements of H3K9me2 levels at the ectopic site (*10X tetO ade6+*) during maintenance (+tetracycline) in different Epe1 mutant backgrounds (N = 2). Error bars represent standard deviations. (E) Live cell imaging of Epe1 and Swi6$^{HP1}$. Four images in each case correspond to DIC, 488 excitation, 560 excitation, and overlay of the three emission channels. mNeonGreen-Epe1 and mCherry- Swi6$^{HP1}$ form co-localized foci in green and red emission channels, respectively (see white arrows). (F) mNeonGreen-Epe1 Y307A fails to form any foci and instead exhibits a diffuse signal that permeates the nucleus. mCherry- Swi6$^{HP1}$ forms foci corresponding to sites of constitutive heterochromatin.

The online version of this article includes the following figure supplement(s) for figure 1:

**Figure supplement 1.** Epe1 mutants are expressed at similar levels to that of the wild-type protein.

in the appearance of red colonies. The sequence-specific initiator, TetR-Clr4-I, dissociates in the presence of tetracycline, enabling us to test whether cells can maintain silencing in the absence of continuous initiation. Wild-type cells are initially red in medium not containing tetracycline (−tetracycline medium), indicating that the reporter gene is initially silenced (establishment). Cells that have a functional copy of Epe1 turn white and exhibit no maintenance when plated on +tetracycline-containing medium. The ability of fission yeast cells to autonomously propagate epigenetic silencing is exquisitely sensitive to Epe1 activity. We observed epigenetic maintenance in cells where Epe1 is either deleted or inactivated, resulting in red or sectored colonies on +tetracycline-containing medium (*Audergon et al., 2015*; *Ragunathan et al., 2015*).

Alanine substitutions of amino acid residues involved in Fe (II) or α-ketoglutarate binding (*epe1 H297A* and *epe1 Y307A*, respectively) disrupt co-factor binding, resulting in the concomitant loss of Epe1 activity. When expressed at endogenous levels, these mutants form red or sectored colonies on +tetracycline-containing medium and resemble *epe1Δ* cells (*Figure 1B*). Replacing the non-conserved tyrosine residue in Epe1 with alanine (*epe1 Y370A*) leads to a similar loss of function phenotype. Hence, despite the lack of conservation, a natural tyrosine substitution within the JmjC domain of Epe1 is essential for its anti-silencing function in cells. We used chromatin immunoprecipitation assays followed by qPCR to measure H3K9me2 levels associated with the reporter gene locus before and after tetracycline addition. Both wild-type and Epe1 mutant strains exhibit high levels of H3K9me2 during establishment (*Figure 1C*). However, wild-type cells lose H3K9me2 approximately 24 h after tetracycline addition. In contrast, Epe1 mutants that exhibit a red or sectored phenotype upon +tetracycline addition retain high levels of H3K9 methylation at the ectopic site (*Figure 1D*). We verified that the expression level of all Epe1 mutant proteins is equal relative to an actin loading control. Hence, neither overexpression artifacts nor changes in protein stability contribute to the maintenance-specific phenotype we observed in our genetic assays (*Figure 1—figure supplement 1B*).

Epe1 is localized at sites of constitutive heterochromatin through its interactions with Swi6$^{HP1}$ (*Ayoub et al., 2003*; *Zofall and Grewal, 2006*). We imaged Epe1 and Swi6$^{HP1}$ in live fission yeast cells using fluorescent protein fusions. We labeled Epe1 with mNeonGreen and Swi6$^{HP1}$ with mCherry. This labeling scheme allows Epe1 and Swi6$^{HP1}$ to be visualized in separate green and red emission channels, respectively. Both fusion proteins were expressed from their endogenous promoters to discount any possible overexpression artifacts. mCherry-Swi6$^{HP1}$ typically exhibits two or three bright foci in individual cells ,corresponding to sites of constitutive heterochromatin (centromeres and telomeres). mNeonGreen-Epe1 co-localizes with mCherry-Swi6$^{HP1}$ as evidenced by the significant overlap between the bright foci that appear green and red emission channels (see white arrows). The overlay also reveals that clusters of Epe1 and Swi6 are co-localized (*Figure 1E*). Surprisingly, an Epe1 co-factor binding mutant, mNeonGreen-Epe1 Y307A, fails to co-localize with mCherry-Swi6$^{HP1}$. Instead, the mutant protein exhibits a diffuse green signal within the nucleus and a complete lack of nuclear foci that co-localize with Swi6$^{HP1}$ (*Figure 1F*). The elevated signal in the cytoplasm could not be attributed to a defect in nuclear localization as cells that do not express any mNeonGreen-Epe1 also exhibit high levels of autofluorescence in the green channel (*Figure 1—figure supplement 1C*). Hence, in addition to affecting any putative enzymatic functions, a co-factor binding mutation within the JmjC domain of Epe1 eliminates protein localization at sites of constitutive heterochromatin.

## Mutations within the JmjC domain disrupt a direct interaction between Epe1 and Swi6[HP1]

We hypothesized that the absence of heterochromatin localization in the Epe1 JmjC mutant could reflect a loss of Swi6[HP1] binding. We used a co-immunoprecipitation assay to compare the interaction between Epe1 and Swi6[HP1] in wild-type and Epe1 mutant cells. We expressed an Epe1-3X FLAG fusion protein at endogenous levels. Using a FLAG antibody, we pulled-down Epe1 and detected its interaction with Swi6[HP1] using a primary antibody. Swi6[HP1] is enriched in pull-down experiments in wild-type cells relative to an untagged control (*Figure 2A*). However, mutations in residues that affect Fe(II) or $\alpha$-ketoglutarate binding (H297A, Y307A, and Y370A) significantly attenuate this interaction (*Figure 2A*). Altering the position of the FLAG epitope tag did not alter the conclusions of our experiments. Epe1 fused to an N-terminal FLAG tag interacts with Swi6[HP1], whereas a mutation within the JmjC domain (H297A) compromises its binding (*Figure 2—figure supplement 1A*).

Epe1 is enriched at sites of constitutive heterochromatin, which include the pericentromeric *dg* and *dh* repeats, the mating type locus, and the telomeres (*Zofall and Grewal, 2006*). We used formaldehyde to crosslink cells followed by chromatin immunoprecipitation to compare heterochromatin occupancy differences between Epe1 wild-type and Epe1 co-factor binding mutants. After crosslinking, we used a FLAG antibody to pull-down the chromatin-bound fraction of Epe1. We used qPCR to measure Epe1 occupancy at the pericentromeric *dg* repeats. Epe1 is enriched within *dg* repeats in wild-type cells and this heterochromatin-specific occupancy pattern is disrupted both in *swi6Δ* and *clr4Δ* cells (*Figure 2B*). Mutations within the JmjC domain that disrupt co-factor binding lead to a substantial reduction in Epe1 occupancy at sites of heterochromatin formation. Consistent with our co-immunoprecipitation studies, all co-factor binding mutants of Epe1 exhibit a significant reduction or completely fail to localize at the pericentromeric *dg* repeats (*Figure 2B*). We altered our fixation conditions using additional reactive crosslinkers and extended the time for formaldehyde crosslinking. Altering crosslinking conditions did not lead to a significant increase in chromatin occupancy amongst Epe1 mutants (*Figure 2—figure supplement 1B*). Based on these results, we concluded that Epe1 co-factor binding mutants exhibit significant defects in their ability to interact with Swi6[HP1] and a complete inability to localize at sites of heterochromatin formation.

Our co-immunoprecipitation, imaging, and ChIP experiments preclude us from making any conclusions as to whether the interaction between Epe1 and Swi6[HP1] is mutation-dependent or requires the putative catalytic functions of Epe1 in vivo. To address this concern, we purified MBP fusions of wild-type Epe1 and Epe1 H297A from insect cells. Swi6[HP1] was purified from *E. coli*. We used TEV protease to cleave the MBP tag and confirmed that recombinant Epe1 remains soluble, but preserved the tag in subsequent purifications for our binding assays (*Figure 2—figure supplement 1C*). We compared the thermal stability of the wild-type and mutant Epe1 protein (Epe1 H297A) using isothermal calorimetry measurements (*Figure 2—figure supplement 1D*). Wild-type Epe1 and Epe1 H297A exhibit similar denaturation temperatures, implying that the mutation within the JmjC domain does not destabilize the protein or cause substantial alterations in protein structure. The difference in peak intensities in the isothermal calorimetry (ITC) profile reflects differences in protein amounts in this assay. These results are consistent with structural studies of JmjC domain-containing proteins where the loss of co-factor binding within the active site does not alter protein structure (*Horton et al., 2011*).

To perform in vitro binding assays, we immobilized Swi6[HP1] on FLAG beads and added three different concentrations of Epe1. Epe1 was detected in these binding assays using an MBP antibody and the total amount of Swi6[HP1] was measured using a FLAG antibody. Through a series of titration measurements, we found that using an MBP antibody and a chemiluminescence based readout produces a very limited linear response, which precludes us from reporting an apparent $K_d$. We verified that the Epe1 protein we purified from insect cells preferentially interacts with Swi6[HP1] as opposed to a second HP1 homolog in *S. pombe*, Chp2[HP1] (*Figure 2C*). Hence, the recombinant Epe1 protein we purified from insect cells recapitulates a known binding preference of Epe1 towards Swi6[HP1] (*Sadaie et al., 2008*). The CSD domain of Swi6[HP1] mediates protein dimerization and regulates Swi6[HP1]-dependent protein-protein interactions (*Canzio et al., 2013*). We expressed and purified a dimerization deficient mutant of Swi6 from *E. coli* (3XFLAG-Swi6[HP1] L315E). Our binding assays using the mutant Swi6[HP1] protein (Swi6[HP1] L315E) reveal a significant reduction in its ability to interact

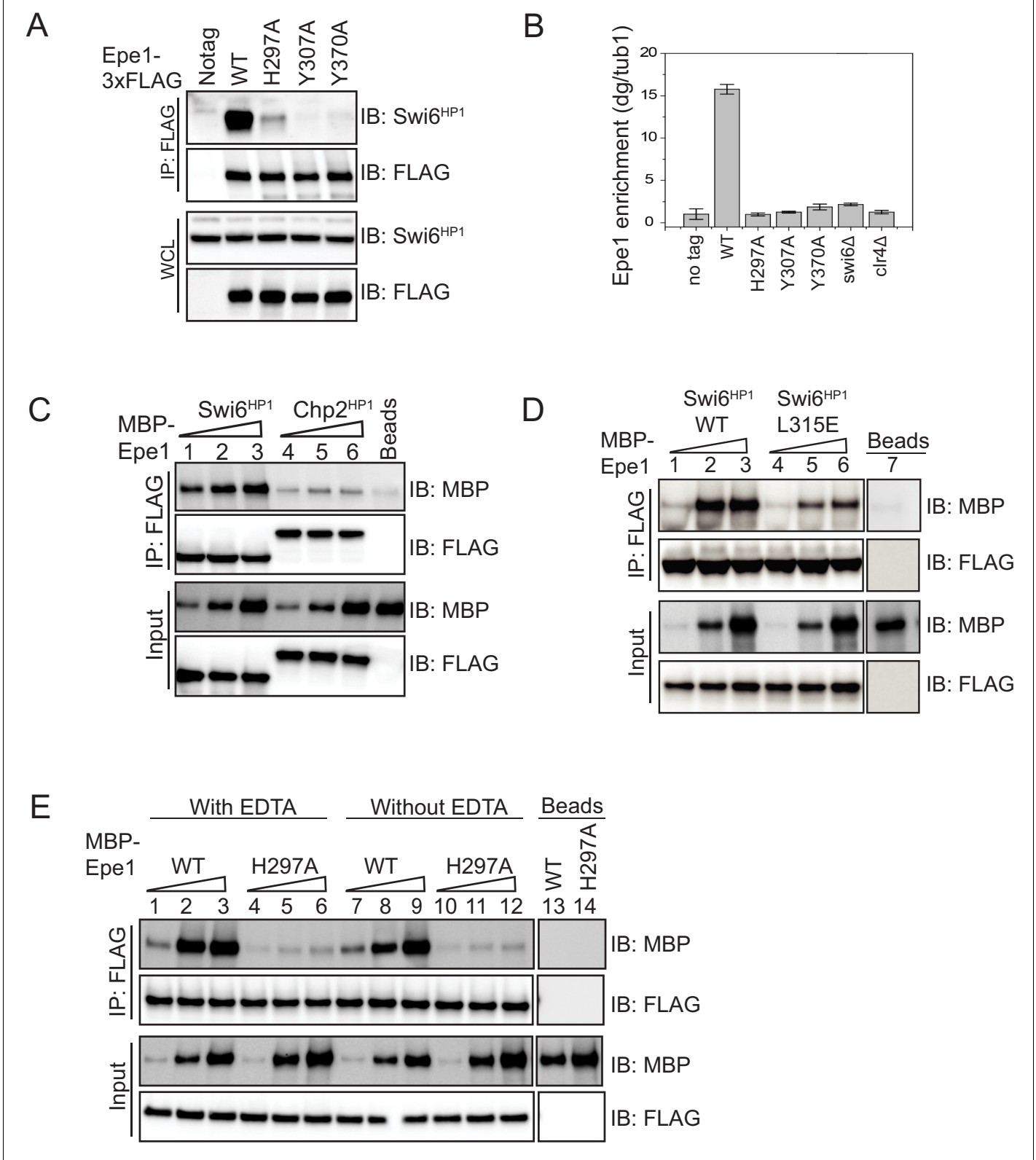

**Figure 2.** Mutations within the JmjC domain perturb a direct interaction between Epe1 and Swi6[HP1]. (**A**) Western blots of co-immunoprecipitation (co-IP) measurements to test the interaction between Epe1-3XFLAG and Swi6[HP1]. Epe1 is detected using a FLAG antibody and Swi6[HP1] is detected using a primary antibody. The interaction between the two proteins is preserved in wild-type cells and is completely eliminated in all Epe1 JmjC mutants. (**B**) ChIP-qPCR measurements of Epe1 occupancy at sites of constitutive heterochromatin (*dg* pericentromeric repeats) (N = 2). Error bars represent

*Figure 2 continued on next page*

*Figure 2 continued*

standard deviations. Epe1 enrichment is reduced to near background levels (*swi6Δ* and *clr4Δ*) in loss of function mutants of Epe1. (**C**) Western blots of in vitro binding assays using recombinant Epe1 protein. Increasing amounts of wild-type MBP-Epe1 protein are added while maintaining a fixed amount of 3X-FLAG-Swi6$^{HP1}$ or 3X-FLAG-Chp2 $^{HP1}$ on beads. Epe1 exhibits a binding preference for Swi6$^{HP1}$over Chp2 $^{HP1}$. (**D**) Increasing amounts of wild-type MBP-Epe1 protein are added while maintaining a fixed amount of 3X-FLAG- Swi6$^{HP1}$ or a CSD domain mutant, 3X-FLAG Swi6$^{HP1}$ L315E on beads. Western blots revealed that a point mutation in the conserved Swi6 $^{HP1}$CSD domain (L315E) leads to reduced levels of interaction between recombinant Epe1 and Swi6$^{HP1}$ L315E. (**E**) Increasing amounts of wild-type MBP-Epe1 and MBP-Epe1 H297A are added while maintaining a fixed amount of 3XFLAG- Swi6$^{HP1}$ on beads. Experiments were performed in the presence and absence of EDTA to measure co-factor independent interactions between the two proteins. The binding recombinant Epe1 H297A binding to Swi6$^{HP1}$ is significantly reduced relative to the wild-type protein.

The online version of this article includes the following figure supplement(s) for figure 2:

**Figure supplement 1.** Characterization of recombinant Epe1 wild-type and Epe1 mutant proteins.
**Figure supplement 2.** Epe1 exhibits no enzymatic activity in the presence or absence of Swi6$^{HP1}$.

with Epe1 (*Figure 2D*). Therefore, Epe1 interacts with Swi6$^{HP1}$ through a conserved mechanism that is shared across other heterochromatin associated factors.

To test whether a mutation within the JmjC domain leads to a loss of interaction between Epe1 and Swi6$^{HP1}$, we compared binding assays between the recombinant wild-type Epe1 protein and the Fe (II) binding deficient mutant, Epe1 H297A. Adding increasing quantities of the wild-type MBP-Epe1 leads to a corresponding increase in the amount of protein that interacts with Swi6$^{HP1}$. However, this type of interaction and increase in binding is not observed in the case of MBP-Epe1 H297A (*Figure 2E*). We compared binding assays performed in the presence and absence of EDTA to rule out any potential contributions that may arise from divalent metals ions that bind to the JmjC domain. The interaction between Epe1 and Swi6$^{HP1}$ is nearly identical in the presence or absence of EDTA (*Figure 2E*). These results suggest that the interaction between Epe1 and Swi6$^{HP1}$ is direct but disrupted by mutations that map to the putative catalytic JmjC domain.

To test whether the enzymatic activity of Epe1 may enhance its interaction with Swi6$^{HP1}$, we added Fe (II), α-ketoglutarate and ascorbate to mimic 'histone-demethylase reaction conditions' in our in vitro binding assays (*Tsukada and Nakayama, 2010*). The addition of co-factors required for histone demethylation did not alter the extent of interaction between Epe1 and Swi6$^{HP1}$ (*Figure 2—figure supplement 1E*). Hence, our in vitro assays fail to capture any effect that co-factor binding itself may have on the interaction between Epe1 and Swi6$^{HP1}$. We also tested whether the interaction between *S. pombe* Swi6$^{HP1}$ and Epe1 might be different from the data we obtained using Swi6$^{HP1}$ purified from *E. coli* (*Figure 2—figure supplement 1F*). Swi6$^{HP1}$ phosphorylation remains intact during our purification as indicated by an upward mobility shift in *S. pombe* Swi6$^{HP1}$ compared to *E. coli* Swi6$^{HP1}$ (*Figure 2—figure supplement 1G*). However, we observed no differences in the interaction profile with Epe1 suggesting that the loss of binding we detected in the Epe1 H297A mutant is independent of Swi6$^{HP1}$ post-translational modifications (*Shimada et al., 2009*). These results suggest that mutations within the JmjC domain of Epe1 may induce a conformational change that attenuates Swi6$^{HP1}$ binding (*Sorida et al., 2019*).

Next, we tested whether Swi6$^{HP1}$ binding to Epe1 may activate its latent enzymatic properties. We performed histone demethylase assays using recombinant Epe1 in the presence and absence of Swi6$^{HP1}$. We used histone H3 tail peptides with a di-methyl or a tri-methyl modification at the lysine nine position (H3K9me2 or H3K9me3 peptides) as substrates. We were unable to detect a mass shift corresponding to the removal of one or more methyl groups in reactions that we performed with Epe1 alone or Epe1 in complex with a five-fold molar excess of Swi6$^{HP1}$ (*Figure 2—figure supplement 2A*). In contrast, JMJD2A, an active demethylase, is fully capable of demethylating an H3K9me3 peptide substrate (*Figure 2—figure supplement 2B*). Hence, Swi6$^{HP1}$ binding to Epe1 is not sufficient to activate its putative enzymatic functions.

## Swi6$^{HP1}$ interacts with the C-terminus of Epe1 through a region that is proximal to the JmjC domain

To map the Swi6$^{HP1}$ interaction site within Epe1, we used an in vitro translation (IVT) assay where we expressed fragments of Epe1 and tested their ability to interact with Swi6$^{HP1}$. We used a computational disorder prediction program to define ordered and disordered regions within the protein

(*Figure 3—figure supplement 1A*). The JmjC domain emerges as one of two ordered regions extending from amino acids 233–434. The second ordered domain that is located within the C-terminus of the protein has no known similarity to existing protein structures and does not have any ascribed function. We designed and expressed partial fragments of Epe1 using rabbit reticulocyte lysates including the full-length protein as a positive control. We added FLAG beads that were pre-incubated with 3XFLAG-Swi6$^{HP1}$ to the IVT extract. A C-terminal fragment of Epe1 spanning 434–948 amino acids and an N-terminal fragment of Epe1 encompassing 1–600 amino acids (Epe1-ΔC) emerged as putative Swi6$^{HP1}$ interaction candidates (*Figure 3—figure supplement 1B*). We hypothesized that the binding region would lie somewhere between amino acid positions 434 and 600. Importantly, this region is proximal to but non-overlapping with the predicted JmjC domain of Epe1.

To validate the conclusions of our IVT binding assay, we expressed and purified two C-terminal fragments of Epe1 from *E. coli.* The first fragment encompasses the entire C-terminus of Epe1 from 434 to 948 amino acids (Epe1$^{434-948}$). The second fragment corresponds to only the minimal Swi6$^{HP1}$ interaction site extending from 434 to 600 amino acids (Epe1$^{434-600}$). We performed binding assays comparing the interaction between the full-length Epe1 protein and the Epe1 C-terminal fragment (Epe1$^{434-948}$) with Swi6$^{HP1}$. The Epe1 C-terminal fragment exhibits an increase in its interaction with Swi6$^{HP1}$ relative to full-length Epe1 (*Figure 3A*). Hence, the C-terminal domain of Epe1, when placed in the context of the full-length protein, is less accessible to Swi6$^{HP1}$. We obtained similar results when we tested the interaction between Epe1$^{434-600}$ and Swi6$^{HP1}$ (*Figure 3—figure supplement 1C*). These observations raise the possibility that the JmjC domain has a steric function and its presence in the context of the full-length protein may impede Swi6$^{HP1}$ binding. We also performed co-immunoprecipitation experiments in cells expressing 3XFLAG-Epe1$^{434-948}$ and detected the same pattern of interaction with Swi6$^{HP1}$ as measured in our in vitro assay (*Figure 3—figure supplement 1D*). We noted that the expression level of the 3X FLAG-Epe1$^{434-948}$ protein expressed from the endogenous *epe1* locus is at least four-fold lower compared to the full-length protein (*Figure 3—figure supplement 1E*).

Although the Swi6$^{HP1}$ binding site lies outside the confines of the JmjC domain of Epe1, point mutations within the putative catalytic JmjC domain perturb a direct interaction between the two proteins (*Figure 2A*). We hypothesized the existence of an interaction in *cis* where the N-terminus half of the protein (Epe1-N, 1–434 amino acids) interacts with its C-terminal portion (Epe1-C, 434–948 amino acids) to interrupt Swi6$^{HP1}$ binding. To test this model, we expressed and purified the N-terminal half of Epe1 containing the JmjC domain (1–434 amino acids) fused to a 3X-FLAG epitope tag in fission yeast cells (3XFLAG-Epe1-N). The purified protein was retained on beads without elution and immediately used for subsequent binding assays. *S. pombe* cells express limiting amounts of the Epe1 N-terminal fragment, which are sufficient for the binding assays described here. We subsequently added a defined amount of the recombinant C-terminal Epe1 fragment (MBP-Epe1$^{434-948}$). We detected a direct interaction between the Epe1-N and Epe1-C terminal fragments (*Figure 3B*, lane 1). Next, we supplemented our binding assays with a two-fold molar excess of recombinant Swi6$^{HP1}$ relative to MBP-Epe1$^{434-948}$. The addition of recombinant Swi6$^{HP1}$ is sufficient to compete with and disrupt a *trans* interaction between the Epe1-N and Epe1-C fragments (*Figure 3B*, lane 2).

We also performed experiments where we used lysates derived from *swi6+* or *swi6Δ* cells expressing an Epe1-N fragment (wild-type or Fe (II) binding mutant allele, Epe1 H297A). We incubated these lysates with a recombinant C-terminal Epe1 fragment, MBP-Epe1$^{434-948}$ immobilized on an amylose resin. Compared to bead-only controls where the amylose resin was incubated with cell lysates, we discovered that the C-terminal Epe1 fragment, MBP-Epe1$^{434-948}$ is able to interact with and pull-down Epe1-N (1-434) in *trans* from a complex mixture of proteins (*Figure 3—figure supplement 1F*). We also observed this *trans* interaction in lysates derived from *swi6Δ* cells further supporting the notion that the interaction between the N and C terminal halves of Epe1 is direct and not mediated by Swi6$^{HP1}$ (*Figure 3—figure supplement 1G*).

One prediction emerging from our biochemical analyses is that expressing the Epe1 C-terminus (Epe1$^{434-948}$) alone might oppose heterochromatin assembly through its direct interaction with Swi6$^{HP1}$. As previously described, we used a reporter gene assay where a TetR-Clr4-I fusion protein initiates heterochromatin establishment in an inducible manner. We expressed Epe1$^{434-948}$ protein in this reporter strain. Surprisingly, this mutant protein, which completely lacks the JmjC domain, can

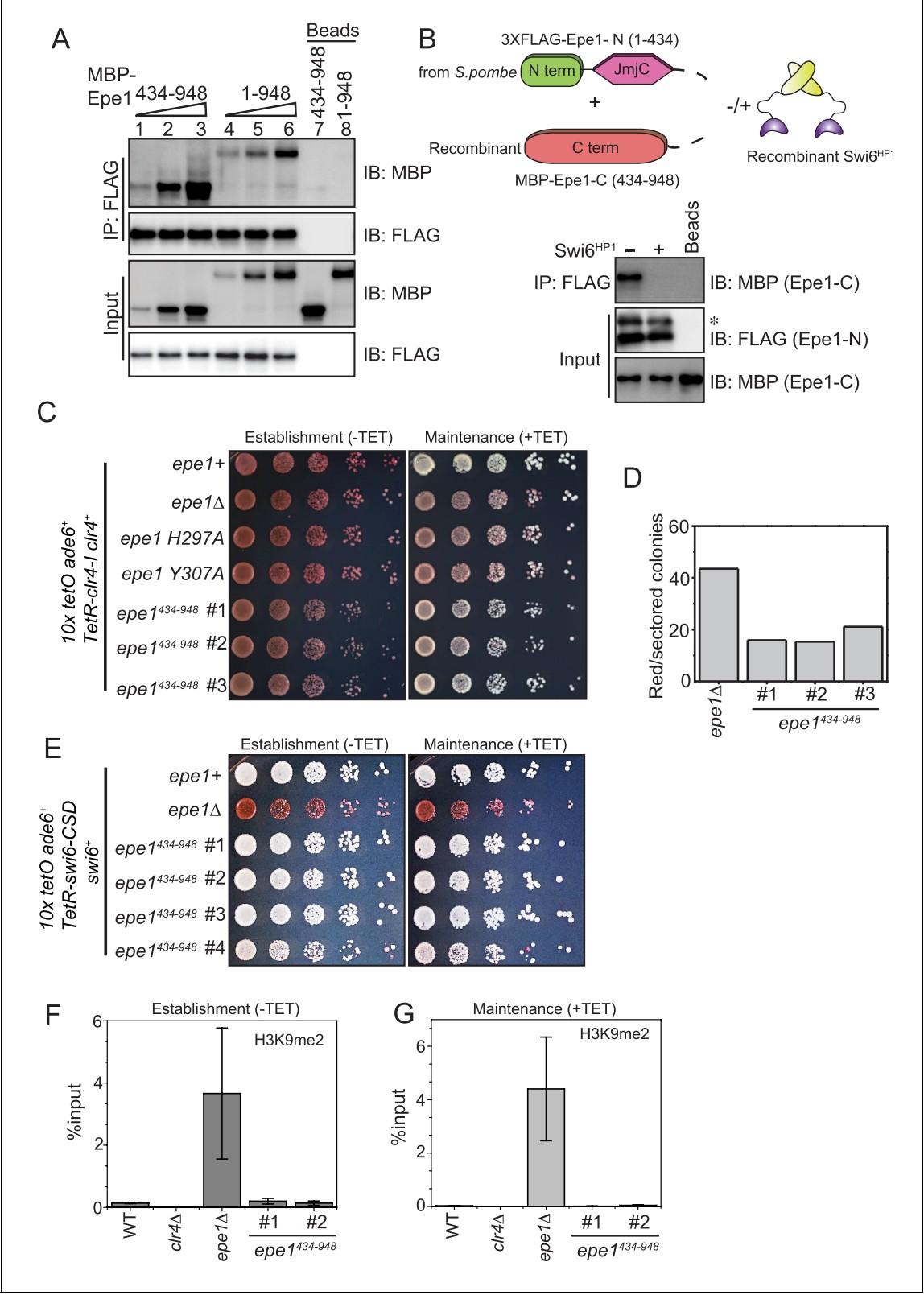

**Figure 3.** Swi6[HP1] interacts with the C-terminus of Epe1 through a region that is proximal to the JmjC domain. (**A**) Western blots of in vitro binding assays to validate the interaction between Swi6[HP1] and the Epe1 C-terminus. Increasing amounts of MBP-Epe1[434-948] protein are added while maintaining a fixed amount of 3X-FLAG-Swi6[HP1] on beads. MBP-Epe1[434-948] exhibits an increase in its interaction with Swi6[HP1] compared to full-length Epe1. (**B**) Western blots of an in vitro binding assay between FLAG-Epe1-N (1–434 amino acids) purified from *S. pombe* and recombinant MBP-Epe1[434-]

*Figure 3 continued on next page*

*Figure 3 continued*

948 in the presence and absence of recombinant Swi6[HP1]. The addition of Swi6[HP1] disrupts a direct *trans* interaction between the N- and C-terminal halves of Epe1. The asterisk in the figure denotes the non-specific FLAG antibody band. (C) Phenotype assays of epigenetic inheritance as a measure of Epe1 activity. Cells expressing Epe1[434-948] were plated on −tetracycline and +tetracycline medium. Cells are initially red during establishment. Despite the absence of the putative catalytic JmjC domain, cells expressing Epe1[434-948] partly turn white on +tetracycline medium. (D) Quantification of red or sectored colonies comparing *epe1Δ* and Epe1[434-948] expressing cells. Epe1[434-948] expressing cells have 30% fewer sectored colonies compared to *epe1Δ* cells. (E) A dominant interaction between the Epe1 C-terminus and Swi6[HP1] opposes heterochromatin establishment. TetR-Swi6-CSD was expressed in cells that harbor *10X TetO* binding sites upstream of an *ade6+* reporter gene. The expression of the Epe1 C-terminus alone in the absence of its putative catalytic JmjC domain disrupts heterochromatin establishment causing cells to remain white under −tetracycline and +tetracycline conditions. (F) ChIP-qPCR measurements of H3K9me2 levels at the ectopic site (*10X tetO ade6+*) during establishment (−tetracycline) in a wild-type, *epe1Δ* and two independent clones expressing Epe1[434-948] (N = 2). Error bars represent standard deviations. (G) ChIP-qPCR measurements of H3K9me2 levels at the ectopic site (*10X tetO ade6+*) during maintenance (+tetracycline) in wild-type, *epe1Δ* and two independent clones expressing Epe1[434-948] (N = 2). Error bars represent standard deviations.

The online version of this article includes the following figure supplement(s) for figure 3:

**Figure supplement 1.** In vitro assays to map the site of interaction between Epe1 and Swi6[HP1].

**Figure supplement 2.** TetR-Swi6 CSD mediated heterochromatin establishment occurs only in the absence of Epe1 and is dependent on H3K9 methylation.

reverse epigenetic maintenance. This reversal leads to a greater proportion of cells that turn white upon +tetracycline addition (*Figure 3C*). The expression of the Epe1 C-terminal fragment, Epe1[434-948] results in a phenotype that is substantially different compared to Epe1 co-factor binding mutants or *epe1Δ* cells. We quantified the number of red or sectored colonies in the Epe1 JmjCΔ mutant compared to *epe1Δ* and wild-type cells. Cells that express a full-length wild-type copy of Epe1 turn white on +tetracycline medium and show no trace of red, pink, or sectored colonies. We observed a three-fold reduction in the number of red or sectored colonies in the Epe1 JmjCΔ mutant compared to Epe1 null cells (*Figure 3D*). Therefore, the Epe1[434-948] mutant is a hypomorphic allele that partially retains wild-type levels of Epe1 anti-silencing activity.

The non-enzymatic function of Epe1 revolves around its dominant mode of interaction with Swi6[HP1]. Therefore, we devised a new ectopic silencing approach where the TetR DNA binding protein was fused to the CSD domain of Swi6[HP1]. We hypothesized that the ability of Epe1 to impede heterochromatin establishment might be more pronounced in this reporter strain. We expressed TetR-Swi6-CSD protein in cells where 10X *TetO* repeats were placed upstream of an *ade6+* reporter. Tethering TetR-Swi6-CSD in wild-type cells fails to establish epigenetic silencing. Cells remain white in the presence or absence of +tetracycline (*Figure 3—figure supplement 2A*). We discovered that Epe1 is the rate-limiting factor that prevents heterochromatin establishment in this ectopic paradigm. Cells turn red and establish epigenetic silencing upon deleting *epe1Δ*. Furthermore, cells remain red even after the addition of +tetracycline consistent with robust sequence-independent epigenetic inheritance in this mutant background (*Figure 3—figure supplement 2A*). Next, we expressed only the C-terminal fragment of Epe1 (Epe1[434-948]). Remarkably, this C-terminal fragment, which is devoid of the JmjC domain, completely blocks heterochromatin establishment. Cells remain white in the presence or absence of +tetracycline (*Figure 3E*). We occasionally observed clonal populations that exhibit red colonies (25% of transformants) consistent with the notion that other domains within Epe1 exert additional enzymatic or non-enzymatic functions (*Bao et al., 2019*; *Sorida et al., 2019*). To test whether the phenotypes we observed depend on H3K9 methylation, we used chromatin immunoprecipitation assays followed by qPCR to measure H3K9me2 levels associated with the *ade6+* reporter gene before and after tetracycline addition. H3K9me2 was observed in cells that turned red in an *epe1Δ* background in the presence or absence of tetracycline (*Figure 3F, G*). Wild-type cells and cells expressing the Epe1-C fragment fail to exhibit any significant enrichment in H3K9me2. In addition, *epe1Δ clr4Δ* cells exhibit a complete loss of epigenetic silencing, further supporting our observations that H3K9 methylation has a causal role in TetR-Swi6-CSD initiated silencing (*Figure 3—figure supplement 2B*).

## H3K9 methylation stimulates complex formation between Epe1 and Swi6[HP1]

We speculated that a heterochromatin-specific trigger might regulate the interaction between the N- and C-terminus of Epe1 to stabilize its interaction with Swi6[HP1]. We expressed Epe1 fused to a 3XFLAG epitope tag in cells that lack the H3K9 methyltransferase, Clr4[Suv39h] or strains where the wild-type H3 allele is replaced with an H3K9R mutant. We performed a co-immunoprecipitation assay where we pull-down Epe1 with a FLAG antibody and measured its interaction with Swi6[HP1]. Although Epe1 interacts with Swi6[HP1] in wild-type cells, this interaction is obliterated in both of the H3K9 methylation deficient mutant strains, *clr4Δ* and H3K9R mutants (*Figure 4A*). Furthermore, deleting histone deacetylases Sir2 or Clr3, both of which affect heterochromatin formation and Swi6[HP1] localization, also resulted in a substantial decrease in the interaction between Epe1 and Swi6[HP1] (*Figure 4B*). In contrast, deleting Mst2, a histone acetyltransferase that enhances hetero-chromatin formation leads to no change in the interaction pattern between Epe1 and Swi6[HP1] (*Figure 4—figure supplement 1A*; *Reddy et al., 2011*). Hence, our results indicate that H3K9 methylation and functional heterochromatin are pre-requisites for complex formation between Epe1 and Swi6[HP1] in cells.

We reconstituted a requirement for H3K9 methylation in stabilizing the interaction between Epe1 and Swi6[HP1] using binding assays as previously described. We supplemented our in vitro binding assays with an unmethylated histone H3 peptide (H3K9me0, H3 1–15 amino acids) or an H3K9 tri-methylated peptide (H3K9me3, H3 1–15 amino acids). Compared to reactions where no-peptide (lanes 1–3) or an unmethylated H3 peptide was added (lanes 4–6), we observed a substantial increase in the interaction between Epe1 and Swi6[HP1] specifically in the presence of an H3K9me3 peptide (lanes 7–9) (*Figure 4C*). An H3K9me2 peptide was also capable of stimulating the interaction between Epe1 and Swi6[HP1] compared to reactions where no peptide was added or assays where an unmodified peptide was used (*Figure 4—figure supplement 1B*). To test whether the stimulation in the interaction between Epe1 and Swi6[HP1] is specific to H3K9 methylation, we carried out binding assays in the presence of an H3K4 tri-methylated peptide (H3K4me3). The addition of an H3K4me3 peptide fails to enhance complex formation between Epe1 and Swi6[HP1], unlike the significant enhancement in binding we observed upon addition of an H3K9me3 peptide (*Figure 4D*). Hence, the stimulatory effect we observed in our binding assays is specific to either H3K9me2 or H3K9me3 peptides.

Our previous results reveal a severe reduction in the interaction between Epe1 H297A and Swi6[HP1] compared to wild-type Epe1 (*Figure 2E*). We compared binding assays between MBP-Epe1 and MBP-Epe1 H297A with Swi6[HP1] in the presence of an H3K9me3 peptide or an H3K9me0 peptide. Although we observed a strong stimulation in the interaction between wild-type Epe1 and Swi6[HP1] (compare lanes 1–3 with 7–9), this stimulatory effect was significantly reduced in the Epe1 H297A mutant (compare lanes 4–6 with 10–12) (*Figure 4—figure supplement 1C*). These results suggest that the JmjC mutant of Epe1 remains refractory to any interaction with Swi6[HP1] in the presence or absence of H3K9 methylation.

One possibility is that Swi6[HP1] undergoes a conformational change that reverses auto-inhibition upon interaction with an H3K9me3 peptide (*Canzio et al., 2013*). To test this hypothesis, we purified a chromodomain mutant Swi6[HP1] protein from *E. coli*. A tryptophan to alanine substitution (W104A) within the Swi6[HP1] chromodomain causes a significant reduction in H3K9 methylation binding (*Jacobs and Khorasanizadeh, 2002*). We used peptide binding assays to confirm that the Swi6[HP1] W104A mutant indeed exhibits a substantial defect in H3K9me3 peptide binding in comparison to the intact Swi6[HP1] protein (*Figure 4—figure supplement 1D*). We tested whether Swi6[HP1] W104A protein binding to Epe1 can also be stimulated in the presence of an H3K9me3 peptide. We added increasing amounts of MBP-Epe1 while maintaining a fixed amount of 3X-FLAG Swi6[HP1] (W104A) on beads. Despite Swi6[HP1] being unable to bind to an H3K9me3 tail peptide, we observed a stimulation in the interaction between Epe1 and the Swi6[HP1] (W104A) (*Figure 4E*).

In addition, we purified a Swi6[HP1] Loop-X mutant that abolishes auto-inhibition and prevents chromodomain-dependent dimerization (*Canzio et al., 2013*). We tested whether the wild-type Epe1 protein and a Swi6[HP1] Loop-X mutant that is constitutively released from auto-inhibition is sensitive to the presence of an H3K9me3 peptide. We purified 3X-FLAG Swi6[HP1] Loop-X mutant from *E. coli* and immobilized the protein on FLAG beads. We added increasing amounts of Epe1 in the

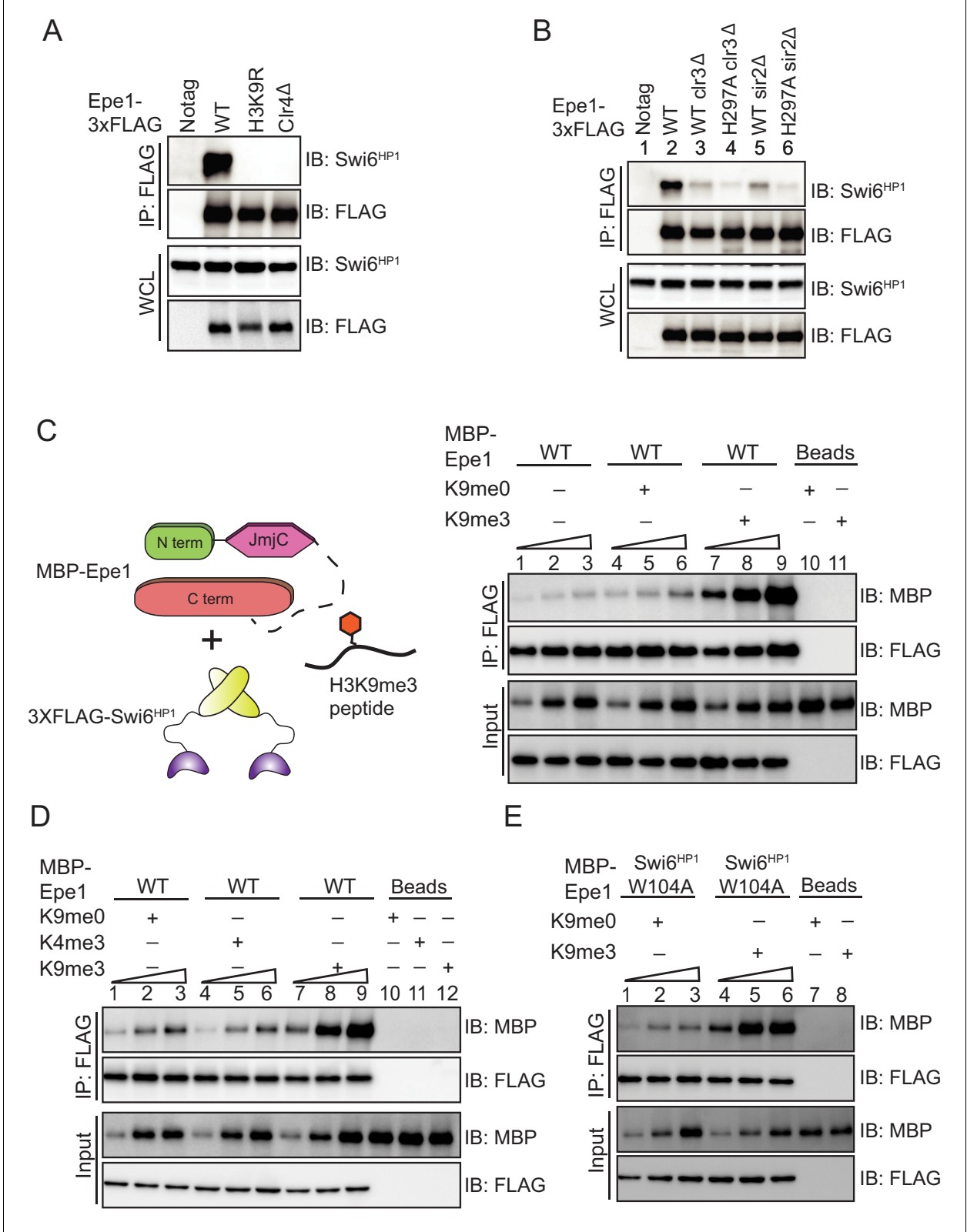

**Figure 4.** H3K9 methylation stimulates complex formation between Epe1 and Swi6HP1. (**A**) Western blots of co-immunoprecipitation measurements reveal an interaction between Epe1-3XFLAG and Swi6HP1 in the case of wild-type cells, which is completely absent in H3K9 methylation deficient cells (*clr4Δ* and H3K9R). (**B**) Co-immunoprecipitation measurements reveal that deletion of histone deacetylases Clr3 or Sir2 which disrupt heterochromatin formation, also leads to a concomitant loss in the interaction between Epe1-3XFLAG and Swi6 HP1. (**C**) Western blots of in vitro binding assays between

*Figure 4 continued on next page*

Figure 4 continued

recombinant Epe1 and Swi6 $^{HP1}$ in the presence of histone H3 tail peptides (1–15 amino acids). Increasing amounts of wild-type MBP-Epe1 are added while maintaining a fixed amount of 3XFLAG- Swi6$^{HP1}$ on beads. Experiments were performed in the presence of an unmethylated H3 peptide (H3K9me0) (lanes 4–6) or an H3K9 tri-methylated peptide (H3K9me3) (lanes 7–9). Epe1 exhibits a significant increase in its ability to interact with Swi6$^{HP1}$ in the presence of an H3K9me3 peptide. (D) Western blots of in vitro binding assays between recombinant Epe1 and Swi6$^{HP1}$ in the presence of differentially methylated histone H3 tail peptides (1–15 or 1–21 amino acids). Increasing amounts of wild-type MBP-Epe1 are added while maintaining a fixed amount of 3XFLAG-Swi6$^{HP1}$ on beads. Experiments were performed in the presence of an unmethylated H3 peptide (H3K9me0), an H3K9 tri-methylated peptide (H3K9me3), or an H3K4 tri-methylated peptide (H3K4me3). Epe1 exhibits a significant increase in its ability to interact with Swi6$^{HP1}$ in the presence of an H3K9me3 but not an H3K4me3 peptide. (E) Western blots of in vitro binding assays between recombinant Epe1 and Swi6$^{HP1}$ W104A in the presence of histone H3 tail peptides (1–15 or 1–21 amino acids). Increasing amounts of wild-type MBP-Epe1 are added while maintaining a fixed amount of 3XFLAG- Swi6$^{HP1}$ W104A on beads. Experiments were performed in the presence of an unmethylated H3 peptide (H3K9me0) or an H3K9 tri-methylated peptide (H3K9me3). Epe1 exhibits an increase in its ability to interact with Swi6$^{HP1}$ W104A in the presence of an H3K9me3 peptide. The online version of this article includes the following figure supplement(s) for figure 4:

**Figure supplement 1.** An Epe1 mutant fails to exhibit H3K9me dependent stimulation in its interaction with Swi6$^{HP1}$.

presence of an H3K9me0 or an H3K9me3 peptide. Although the H3K9me0 peptide has no effect on the interaction between Epe1 and Swi6$^{HP1}$ Loop-X, the presence of an H3K9me3 peptide substantially enhances their interaction. Therefore, Epe1 remains responsive to the presence of an H3K9 methylated peptide even in a context where Swi6$^{HP1}$ is constitutively released from auto-inhibition (*Figure 4—figure supplement 1E*).

## An Epe1 C-terminal truncation mutant exhibits enhanced binding to Swi6$^{HP1}$ and is insensitive to H3K9 methylation

Our previous results suggest that Epe1 might have a latent capacity to bind to H3K9 methylated peptides or histones. We tested whether Epe1 directly binds to an H3K9me3 peptide and specifically interacts with H3K9 methylated histones. We performed peptide binding assays where a biotinylated H3K9me3 peptide was immobilized on streptavidin beads. Our binding assays detect a direct interaction between Epe1 and an H3K9me3 peptide as opposed to an unmodified H3K9me0 peptide. Furthermore, Epe1 selectively interacts with H3K9 methylated histones as opposed to H3K4 methylated histones. (*Figure 5A,B*). Next, we expressed and purified a C-terminal truncation mutant of Epe1, MBP-Epe1-ΔC from Sf9 insect cells, which includes amino acids 1–600 and includes the putative catalytic JmjC domain. We found that Epe1-ΔC can also directly bind to an H3K9me3 peptide and specifically interacts with H3K9 methylated histones (*Figure 5—figure supplement 1A, B*). Based on these observations, we hypothesize that the JmjC domain of Epe1 (amino acids 233–434) might be primarily responsible for H3K9 methylation recognition and binding.

We previously demonstrated that the addition of Swi6$^{HP1}$ disrupts an interaction in *trans* between the Epe1-N and Epe1-C terminus (*Figure 3B*). The stimulation in Swi6$^{HP1}$ binding that we observed prompted us to test whether Epe1 binding to an H3K9 methylated peptide might also have a similar function and interrupt a *trans* interaction between the Epe1 N- and C- terminal fragments. Disrupting their interaction would enable Swi6$^{HP1}$ to gain access to the C-terminus of Epe1. To test this model, we measured a *trans* interaction between the Epe1 N- and C-terminal halves in the presence of an H3K9me0 peptide or an H3K9me3 peptide (*Figure 5C*). We purified an Epe1-N fragment fused to a 3X FLAG epitope tag. Next, we added a recombinant Epe1 C-terminal fragment (434–948 amino acids) in the presence of an H3K9me0 peptide or an H3K9me3 peptide. We observed an interaction in *trans* between the N- and C-terminal halves of Epe1 in binding assays with no peptide or an H3K9me0 peptide. However, the addition of an H3K9me3 peptide eliminates the interaction between the N- and C-terminal fragments of Epe1 (*Figure 5C*).

Based on our observations, we hypothesized that the C-terminus of Epe1 might have a regulatory function in enforcing an H3K9 methylation-dependent mode of interaction between Epe1 and Swi6$^{HP1}$. To test this model, we expressed a C-terminal truncation of Epe1 (Epe1-ΔC) fused to a 3XFLAG epitope tag in fission yeast cells. We performed a co-IP experiment to test the interaction between Epe1-ΔC, and Swi6$^{HP1}$ compared to the full-length protein. We also expressed an N-terminal fragment of Epe1 (Epe1-N) which lacks the Swi6$^{HP1}$ binding site. Our co-IP assays detect a substantial increase in the interaction between Epe1-ΔC and Swi6$^{HP1}$, compared to a weak interaction in the case of the full-length protein and no interaction in the case of Epe1-N (*Figure 5—figure*

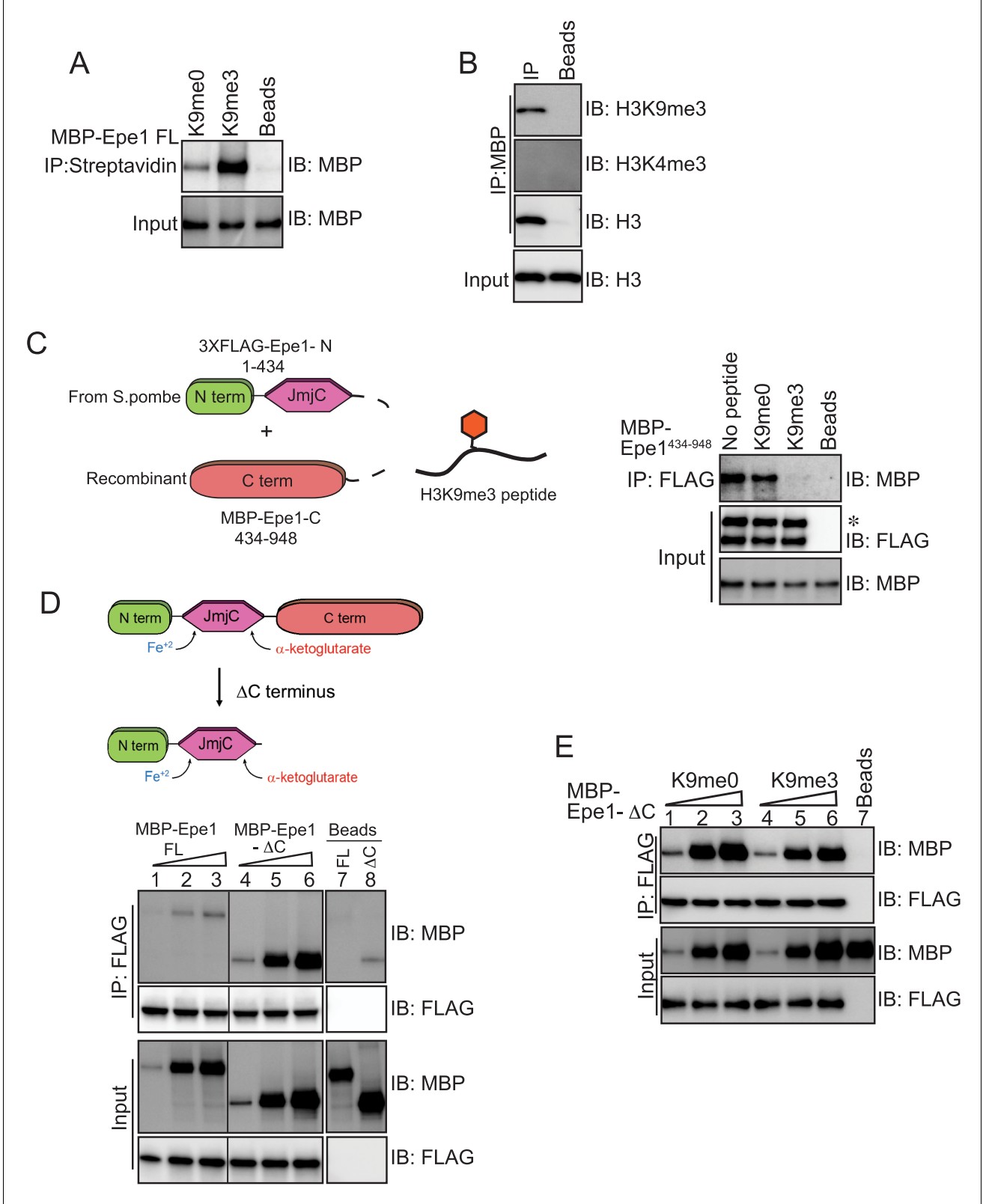

**Figure 5.** An Epe1 C-terminal truncation mutant exhibits enhanced binding to Swi6 and is insensitive to H3K9 methylation. (**A**) Biotinylated peptides (H3K9me3 or H3K9me0) are immobilized on streptavidin beads and incubated with MBP-Epe1. MBP-Epe1 preferentially associates with an H3K9me3 peptide compared to an H3K9me0 peptide. (**B**) Epe1 specifically interacts with H3K9me3 histones. Calf-thymus histones containing an unspecified mixture of differentially modified histones were incubated with MBP-Epe1 immobilized on an amylose resin. Epe1 preferentially interacts with H3K9

*Figure 5 continued on next page*

Figure 5 continued

methylated histones as opposed to H3K4 methylated histones. (C) Western blots of an in vitro binding assay to determine whether an H3K9me3 peptide disrupts a *trans* interaction between the N- and C-terminal halves of Epe1. The N-terminus of Epe1, 3X FLAG-Epe1-N (1-434) was purified using a FLAG antibody from fission yeast cells. The purified protein was incubated with recombinant MBP-Epe1$^{434-948}$ protein in the presence of an H3K9me0 peptide and an H3K9me3 peptide. The H3K9me3 peptide specifically disrupts a direct *trans* interaction between the N- and C-terminal halves of Epe1. (D) Western blots of in vitro binding assays comparing the interactions between recombinant Epe1-ΔC and Swi6$^{HP1}$. Increasing amounts of wild-type MBP-Epe1 or MBP-Epe1-ΔC are added while maintaining a fixed amount of 3XFLAG- Swi6$^{HP1}$ on beads. Epe1-ΔC (lanes 4–6) clearly surpasses the full-length protein (lanes 1–3) in terms of its ability to interact with Swi6$^{HP1}$. (E) Western blots of in vitro binding assays between recombinant Epe1-ΔC and Swi6$^{HP1}$ in the presence of histone H3 tail peptides (1–15 amino acids). Increasing amounts of MBP-Epe1-ΔC are added while maintaining a fixed amount of 3XFLAG-Swi6$^{HP1}$ on beads. Experiments were performed in the presence of an unmethylated H3 peptide (H3K9me0) (lanes 1–3) or an H3K9 tri-methylated peptide (H3K9me3) (lanes 4–6). Epe1-ΔC exhibits no change in its interaction with Swi6$^{HP1}$ in the presence of an H3K9me3 peptide.

The online version of this article includes the following figure supplement(s) for figure 5:

**Figure supplement 1.** An Epe1 C-terminal truncation mutant interacts with Swi6$^{HP1}$ in the absence of H3K9 methylation in cells.

*supplement 1C*). Next, we performed in vitro binding assays to test the extent of interaction between recombinant MBP-Epe1-ΔC and FLAG-Swi6$^{HP1}$ relative to the full-length Epe1 protein. Epe1-ΔC significantly outperforms the full-length protein in its ability to interact with Swi6$^{HP1}$ (compare lanes 4–6 with lanes 1–3) (*Figure 5D*).

We performed binding assays where we added increasing amounts of Epe1-ΔC while maintaining a fixed concentration of Swi6$^{HP1}$ on beads in the presence of an H3K9me0 or an H3K9me3 peptide. Surprisingly, the addition of a modified peptide had no effect on the interaction between Swi6$^{HP1}$ and Epe1-ΔC (*Figure 5E*). This is not because of saturation of the chemiluminescent signal as the addition of increasing amounts of Epe1-ΔC continues to produce a concomitant increase in its interaction with Swi6$^{HP1}$. Previously, we demonstrated that the interaction between full-length Epe1 and Swi6$^{HP1}$ requires functional heterochromatin. We performed a co-IP measurement in *clr4+* and *clr4Δ* cells expressing FLAG-Epe1-ΔC. Consistent with our in vitro binding assays and unlike the full-length Epe1 protein, we observed an interaction between Epe1-ΔC and Swi6$^{HP1}$ in both *clr4+* and *clr4Δ* cells (*Figure 5—figure supplement 1D*). Therefore, Epe1-ΔC remains unresponsive to the presence of H3K9 methylation and does not undergo any further stimulation in its ability to interact with Swi6$^{HP1}$.

## Epe1 inhibits Swi6-dependent heterochromatin assembly through a non-enzymatic process

We hypothesized that Epe1 might outcompete other heterochromatin associated proteins that localize to sites of heterochromatin formation through a dominant interaction with Swi6$^{HP1}$. Genetic studies reveal that Epe1 and a histone deacetylase Clr3 have opposing effects on nucleosome turnover (*Aygün et al., 2013*). One possibility is that the interaction between Epe1 and Swi6$^{HP1}$ excludes heterochromatin agonists, such as Clr3 from sites of heterochromatin formation. We used a co-immunoprecipitation assay to measure the extent of interaction between Clr3 and Swi6$^{HP1}$. Clr3 was fused to a 3X V5 epitope tag and expressed in a wild-type Epe1 and an Epe1 H297A background. We used a V5 antibody to pull-down Clr3 , after which we detected Swi6$^{HP1}$ using a primary antibody. We measured a weak interaction between Swi6$^{HP1}$ and Clr3 in a wild-type Epe1 background that substantially increases in an Epe1 H297A strain background (*Figure 6A*). This positive change in the interaction between Clr3 and Swi6$^{HP1}$ occurs in the absence of any increase in Swi6$^{HP1}$ occupancy at the pericentromeric repeats (*Figure 6B*). Although Clr3 is also recruited to sites of heterochromatin formation by interacting with Chp2$^{HP1}$, the levels of Chp2$^{HP1}$ in fission yeast cells are approximately 100-fold lower compared to Swi6$^{HP1}$ (*Sadaie et al., 2008*). This difference in stoichiometry between Swi6$^{HP1}$ and Chp2$^{HP1}$ could explain how Epe1 may have an outsized role in interfering with heterochromatin assembly by selectively disrupting Swi6$^{HP1}$ associated protein complexes.

We also measured Epe1 localization in *clr3Δ* mutant cells. Consistent with our model, we observed a three-fold increase in Epe1 localization at the pericentromeric *dg* repeats and the mating type locus (*mat*) (*Figure 6C, D*). In contrast, the deletionof *clr3Δ* results in the complete loss of Epe1 localization at the telomeres (*Figure 6E*).

We hypothesized that directly tethering Clr3 at an ectopic site would eliminate the requirement for Swi6$^{HP1}$ for its recruitment at sites of heterochromatin formation. In this genetic context, we

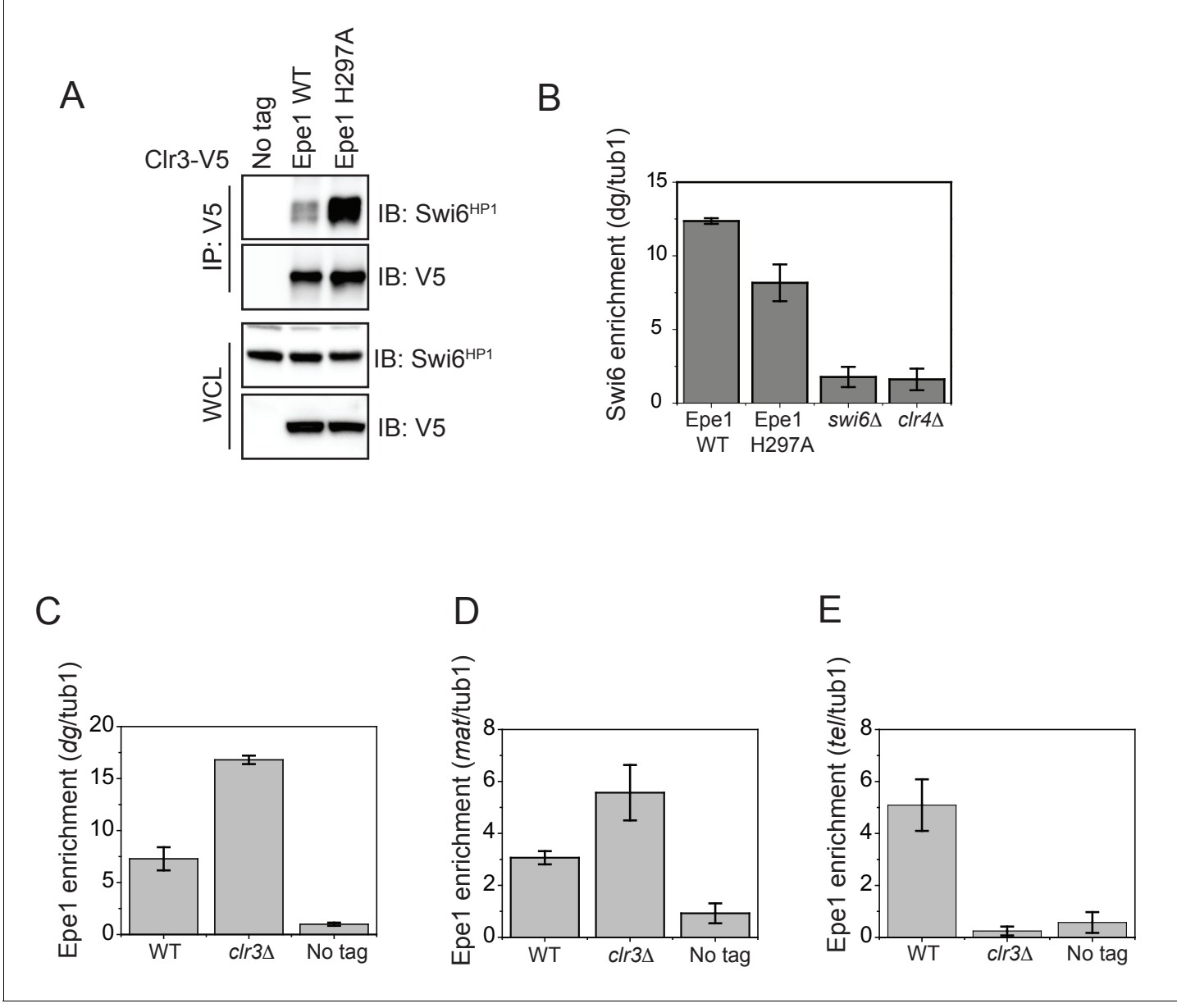

**Figure 6.** The interaction between Epe1 and Swi6 displaces Clr3 from localizing to sites of heterochromatin formation. (**A**) Epe1 displaces Clr3 from sites of heterochromatin formation. Western blots of a co-immunoprecipitation experiment reveal that Clr3-3X V5 interacts weakly with Swi6[HP1] in a wild-type Epe1 background. This interaction increases significantly in an Epe1 H297A background. (**B**) ChIP-qPCR measurements of Swi6[HP1] occupancy at sites of constitutive heterochromatin in wild-type and Epe1 mutant cells. The increase in interaction between Swi6[HP1] and Clr3 is independent of Swi6[HP1] levels at constitutive heterochromatin. (**C**) ChIP-qPCR measurements of Epe1 occupancy at *dg* pericentromeric repeats in *clr3Δ* cells (N = 2). Error bars represent standard deviations. Epe1 enrichment is increased two to three-fold in cells at *dg* in *clr3Δ* strains. (**D**) ChIP-qPCR measurements of Epe1 occupancy at the mating-type locus (mat) in *clr3Δ* cells (N = 2). Error bars represent standard deviations. Epe1 enrichment is increased two to three-fold at the *mat locus* in *clr3Δ* strains. (**E**) ChIP-qPCR measurements of Epe1 occupancy at telomeres (*tlh1*) in *clr3Δ* cells (N = 2). Error bars represent standard deviations. Epe1 localization is completely eliminated at the telomeres in *clr3Δ* strains.

rationalized that Epe1 would be unlikely to block Clr3 recruitment. One caveat of this experimental strategy is that it does not rule out the possibility that Epe1 and Clr3 have antagonistic catalytic functions. However, this experiment directly tests whether the Clr3 level itself might be rate-limiting at sites of heterochromatin formation. To test this model, we engineered synthetic heterochromatin domains where *10X Tet operator* (*10XTetO*) and *10X Gal4 DNA binding sites* (*10X Gal4*) are placed next to each other and inserted upstream of an *ade6+* reporter gene. This enables inducible

heterochromatin formation via TetR-Clr4-I recruitment while the Gal4 DNA binding domain allows for the orthogonal recruitment of additional chromatin effectors to the ectopic site (*Figure 7A*). In the absence of any additional chromatin modifiers, TetR-Clr4-I recruitment to the newly engineered ectopic site (*10X Gal4-10X TetO- ade6+*) leads to the appearance of red colonies on –tetracycline medium. Upon the addition of tetracycline, cells turn white consistent with a causal role for Epe1 in resetting epigenetic inheritance (*Figure 1B*). We fused the Gal4 DNA binding domain (Gal4 DBD) to two histone deacetylases, Clr3 (a class II histone deacetylase) and Sir2 (a NAD-dependent histone deacetylase) enabling these proteins to be constitutively tethered at the Gal4 DNA binding sites. The TetR-Clr4-I fusion initiates heterochromatin formation and cells turn red on medium lacking tetracycline in the presence of Gal4-Clr3 or Gal4-Sir2 (*Figure 7A*). However, upon exposure to tetracycline, the constitutive tethering of Clr3 but not Sir2 promotes epigenetic inheritance. Cells exhibit a red and sectored phenotype in cells where Clr3 is artificially tethered, despite Epe1 still being present (*Figure 7B*). The phenotypes in cells where Gal4-Clr3 is tethered are remarkably similar to cells that lack Epe1 (*epe1Δ*). We measured changes in H3K9me2 levels associated with the *ade6+* reporter gene using ChIP-qPCR. Cells maintain high H3K9me2 levels in the presence of Gal4-Clr3 before and after tetracycline (*Figure 7C-D*). These results suggest that constitutively tethering Clr3 to sites of heterochromatin formation is sufficient to oppose Epe1 activity resulting in maintenance of H3K9 methylation even after +tetracycline addition.

Tethering Gal4-Clr3 in the absence of TetR-Clr4-I causes no change in reporter gene silencing or H3K9me2 levels at the ectopic site (*Figure 7—figure supplement 1A,B,C*). Hence, Gal4-Clr3 cannot initiate H3K9 methylation de novo in fission yeast (*Figure 7—figure supplement 1B,C*). This lack of de novo silencing is consistent with the notion that HDAC proteins in fission yeast collaborate with H3K9 methyltransferases to establish epigenetic silencing. Furthermore, expressing Clr3 minus the Gal4 DBD fusion (Clr3 ΔGal4) leads to a loss of epigenetic maintenance on +tetracycline-containing medium. Therefore, Clr3 must be recruited in *cis* to oppose the anti-silencing effects of Epe1 (*Figure 7—figure supplement 1A*). Consistent with the phenotypes that we observed, H3K9me2 levels are high during establishment but completely absent during maintenance in cells expressing diffusible Clr3 protein (*Figure 7—figure supplement 1B,C*). Hence, it is the sequence-specific recruitment of Clr3, rather than protein dosage, that facilitates H3K9 methylation maintenance. Therefore, Clr3 recruitment in *cis* is required to maintain silent epigenetic states and oppose Epe1 activity. This property of heterochromatin maintenance is RNAi independent as cells continue to exhibit a red or sectored appearance in a Dicer deficient background (*dcr1Δ*) (*Figure 7—figure supplement 1D*).

The read-write activity of Clr4$^{Suv39h}$ is essential for the inheritance of silent epigenetic states in a sequence-independent manner (*Audergon et al., 2015*; *Ragunathan et al., 2015*). This H3K9 methylation-dependent positive feedback loop is disrupted in a Clr4$^{Suv39h}$ chromodomain mutant (*Zhang et al., 2008*). To test whether the chromodomain is essential for maintenance when Clr3 is tethered, we replaced the wild-type allele of Clr4$^{Suv39}$ with a Clr4 mutant that lacks the chromodomain (*clr4ΔCD*). Cells that are initially red in –tetracycline medium turn white on +tetracycline medium in a *clr4ΔCD* expressing mutant (*Figure 7E*). H3K9me2 levels in *clr4ΔCD* mutants are similar to those of wild-type cells during establishment. However, H3K9 methylation is absent upon +tetracycline addition in *clr4ΔCD* expressing strains. Hence, the inheritance of H3K9 methylation depends on the read-write activity of Clr4$^{Suv39h}$ despite Clr3 being constitutively tethered (*Figure 7F, G*).

## Discussion

The establishment and maintenance of epigenetic states is primarily thought to depend on a balance of enzymatic activities between readers, writers, and erasers of histone modifications (*Allis and Jenuwein, 2016*). However, the replication-dependent and independent turnover of histones serve as a major mechanism that shapes genome-wide patterns of histone methylation (*Chory et al., 2019*). In *Drosophila*, modified histones are turned over more than once during each cell-cycle , which limits their capacity to serve as carriers of epigenetic information (*Coleman and Struhl, 2017*; *Deal et al., 2010*; *Laprell et al., 2017*). In principle, passive genome-wide nucleosome exchange can compete with histone modification-dependent read-write mechanisms to oppose epigenetic inheritance.

Our data support the notion that enzymatic erasure is at least in part, dispensable for regulating the inheritance of transcriptionally silent epigenetic states in fission yeast. Our observations are fully

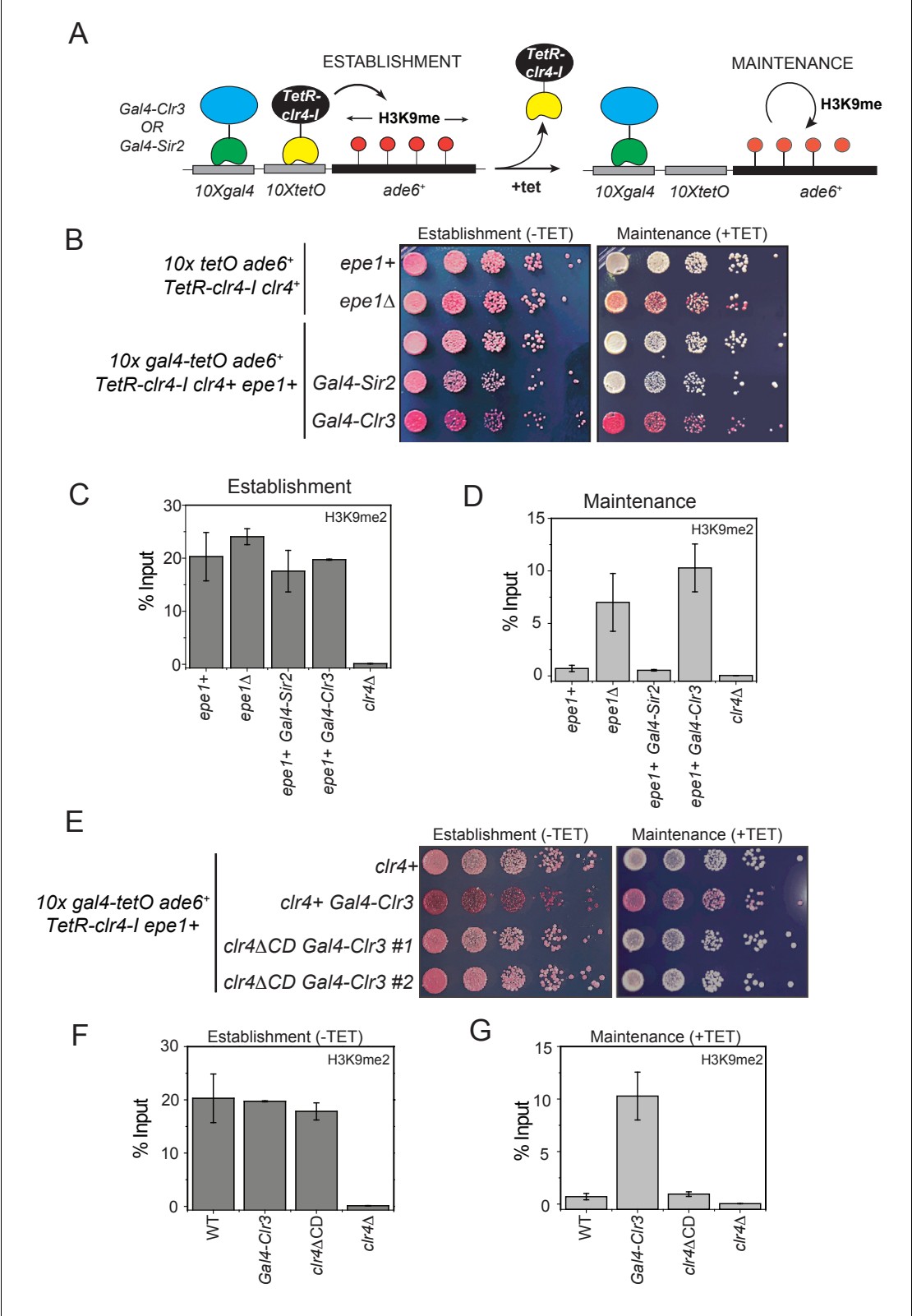

**Figure 7.** H3K9 methylation regulates the non-enzymatic functions of a putative histone demethylase Epe1, which interacts weakly with Swi6[HP1]. (**A**) A reporter system to detect epigenetic inheritance of H3K9 methylation in the presence of orthogonal chromatin effectors. Heterochromatin initiation depends on TetR-Clr4-I binding (-tetracycline). Orthogonal chromatin effectors can be recruited to the *10xgal4* DNA binding site via a Gal4 DNA binding domain (Gal4 DBD). The addition of tetracycline promotes TetR-Clr4-I dissociation to measure epigenetic inheritance in the presence of an

*Figure 7 continued*

orthogonal chromatin effector. (**B**) A color-based assay to detect the establishment and maintenance of epigenetic states. The establishment of epigenetic silencing (-tetracycline) results in red colonies. Tethering Gal4-Clr3 but not Gal4-Sir2 at an ectopic site promotes epigenetic inheritance of H3K9 methylation resulting red or sectored colonies in +tetracycline containing medium in *epe1+* (**C**) ChIP-qPCR measurements of H3K9me2 levels at the ectopic site (*10X tetO ade6+*) during establishment (-tetracycline). (N=2). Error bars represent standard deviations. (**D**) ChIP-qPCR measurements of H3K9me2 levels at the ectopic site (*10X tetO ade6+*) during maintenance (+tetracycline) (N=2). Error bars represent standard deviations. (**E**) The deletion of the Clr4 chromodomain (*clr4ΔCD*) inactivates read-write functions which affects epigenetic inheritance. Cells that are initially red during heterochromatin establishment (-tetracycline) turn white during maintenance (+tetracycline) in a *clr4ΔCD* background. (**F**) ChIP-qPCR measurements of H3K9me2 levels at the ectopic site (*10X tetO ade6+*) during establishment (-tetracycline) in *clr4ΔCD* mutant cells (N=2). Error bars represent standard deviations. ChIP-qPCR values from Figure 7C are plotted as a reference. (**G**) ChIP-qPCR measurements of H3K9me2 levels at the ectopic site (*10X tetO ade6+*) during maintenance (+tetracycline) in *clr4ΔCD* mutant cells (N=2). Error bars represent standard deviations. Error bars represent standard deviations. ChIP-qPCR values from Figure 7D are plotted as a reference.

The online version of this article includes the following figure supplement(s) for figure 7:

**Figure supplement 1.** Clr3 tethering in cis and the read-write activity of Clr4 is required for epigenetic inheritance.

consistent with earlier work demonstrating an antagonistic relationship between Epe1, which enhances nucleosome turnover at sites of heterochromatin formation, and Clr3, which suppresses this process (*Aygün et al., 2013*). In addition, Epe1 also has newly defined roles in recruiting members of the SAGA complex which mediate histone acetylation, a chromatin feature that is also associated with increased nucleosome turnover (*Bao et al., 2019*). Recent work demonstrated a surprising and unexpected function for the N-terminus of Epe1 in transcriptional activation, which suppresses stochastic, epigenetic silencing (*Sorida et al., 2019*). Our findings are complementary to these recent studies and expand the gamut of anti-silencing functions associated with Epe1.

Auto-inhibition is a widely used strategy that proteins use to activate their latent functions in response to specific cellular or environmental signals (*Pufall and Graves, 2002*). The enzymatic and non-enzymatic properties of proteins can be regulated by internal control mechanisms that act in *cis*. For example, in the case of the chromatin remodeler ALC1, poly ADP ribosylation in response to DNA damage releases the macrodomain of the protein from an auto-inhibited state and activates its latent ATPase-dependent nucleosome remodeling function (*Lehmann et al., 2017*; *Singh et al., 2017*). The non-catalytic heterochromatin associated protein Swi6$^{HP1}$ is auto-inhibited by a histone H3 mimic sequence (*Canzio et al., 2013*). Release from auto-inhibition switches the protein to a spreading competent state.

We discovered that a weak interaction between Epe1 and Swi6$^{HP1}$ can be bolstered by H3K9 methylation. We propose that a *cis* interaction between the N- and C-terminal halves of Epe1 opposes Swi6$^{HP1}$ binding. These results suggest that Epe1 may be preserved in an auto-inhibited state until it interacts with H3K9 methylation (*Figure 8*). In the absence of H3K9 methylation, the two proteins fail to form a stable complex in vivo despite their ability to interact directly with each other in vitro. The ability of Epe1 to bind to H3K9 methylation raises the possibility that it competes with Swi6$^{HP1}$ for a shared binding site. However, heterochromatin consists of dense networks of H3K9 methylated nucleosomes. It is also possible that Epe1 while interacting with Swi6$^{HP1}$, can also be stimulated via an interaction with an adjacent or a distal nucleosome. Our data do not lend direct support to a structural change that occurs within Epe1 upon H3K9me3 peptide binding but strongly suggest that modified histones can influence the stability of chromatin associated complexes in living cells.

Swi6$^{HP1}$ is dynamic and undergoes rapid exchange on the millisecond timescale between the free and H3K9 methylation-bound state (*Cheutin et al., 2004*; *Cheutin et al., 2003*). Given the relative abundance of Swi6$^{HP1}$ in cells, a simple protein-protein interaction between Epe1 and Swi6$^{HP1}$ would effectively titrate Epe1 away from sites of heterochromatin formation. By enforcing an H3K9 methylation-dependent mode of interaction, Epe1 selectively targets a sub-population of Swi6$^{HP1}$ molecules that are bound to sites of H3K9 methylation and have a causal role in heterochromatin formation. Therefore, one outcome of this H3K9 methylation-dependent mode of interaction is that Epe1 selectively interacts with heterochromatin-bound Swi6$^{HP1}$ molecules as opposed to freely diffusing Swi6$^{HP1}$ proteins. It is likely that this balance of protein-protein interactions is tunable via post-translational modifications. For example, Swi6$^{HP1}$ phosphorylation compromises Epe1 binding and promotes histone deacetylase recruitment to sites of heterochromatin formation

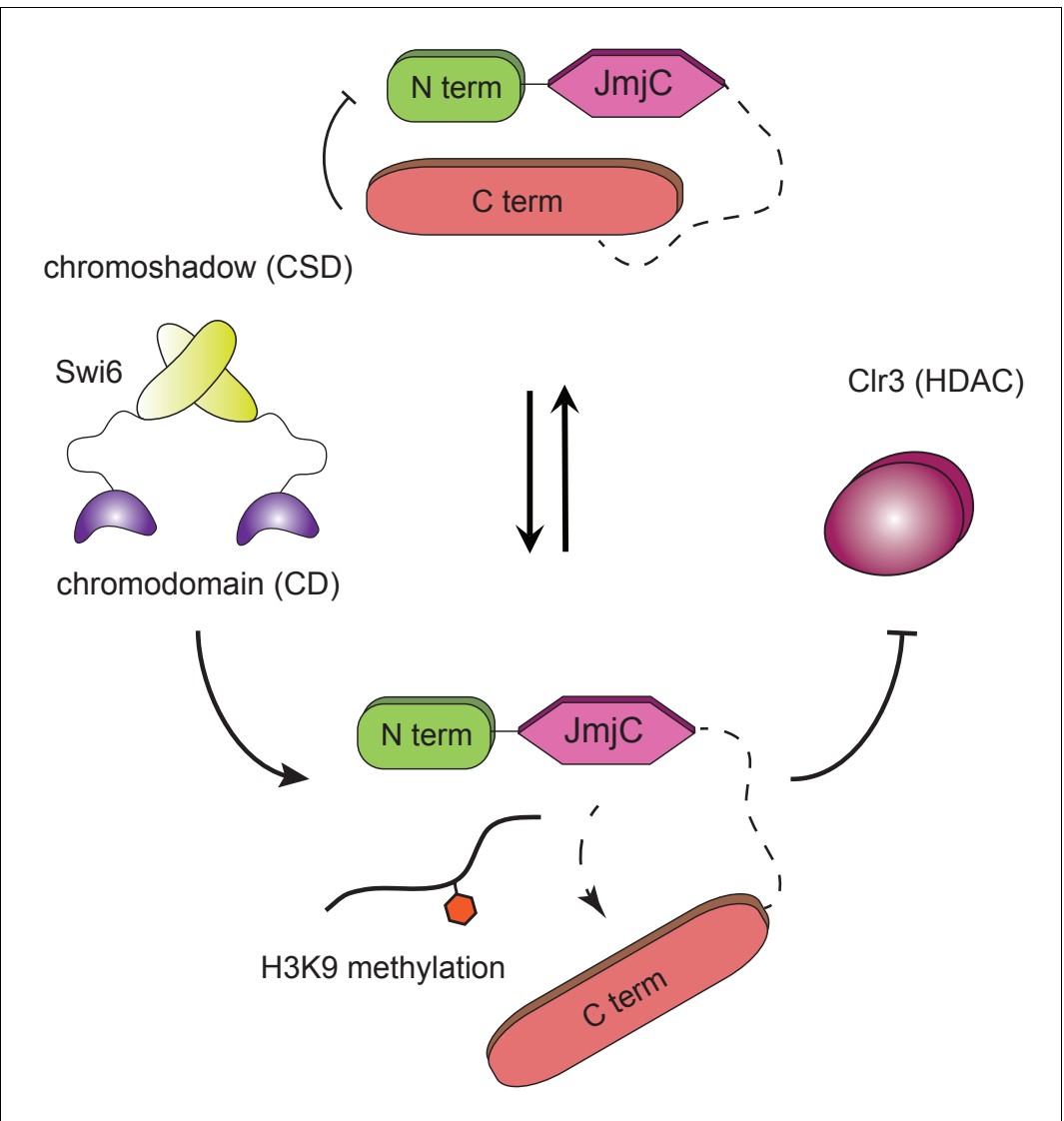

**Figure 8.** H3K9 methylation regulates the non-enzymatic functions of a putative histone demethylase. Epe1 interacts weakly with Swi6[HP1]. Point mutations within the catalytic JmjC domain of Epe1 disrupts its interaction with Swi6[HP1]. H3K9 methylation binding stimulates the interaction between Epe1 and Swi6[HP1] which stabilizes complex formation at sites of heterochromatin formation. The Epe1 C-terminus enforces an H3K9 methylation dependent mode of interaction between Epe1 and Swi6[HP1] which displaces the histone deacetylase, Clr3 from sites of heterochromatin formation. Our model reveals how H3K9 methylation stabilizes an inhibitory complex consisting of Epe1 and Swi6[HP1] and suggests how a non-enzymatic function associated with Epe1 C-terminus regulates epigenetic inheritance.

(*Shimada et al., 2009*). Our studies demonstrate that altering the balance of Swi6[HP1]-dependent protein-protein interactions profoundly affects the stability and heritability of silent epigenetic states.

Tethering Clr3 at an ectopic site renders heterochromatin refractory to the anti-silencing effects of Epe1. Our observations are in part, similar to previous findings relating to the fission yeast mating type locus where DNA binding proteins, Atf1 and Pcr1 recruit Clr3 to maintain epigenetic silencing following an RNAi-dependent initiation mechanism (*Wang and Moazed, 2017*; *Yamada et al., 2005*). We also recapitulate a unique requirement for the sequence-dependent recruitment of HDAC proteins in *cis* to facilitate the inheritance of silent epigenetic states despite the presence of anti-silencing factors such as Epe1. We favor a model where histone deacetylases such as Clr3 create

a chromatin environment that promotes the read-write enzymatic function of Clr4[Suv39h]. In essence, Epe1 prevents Clr3 mediated histone hypoacetylation by forming an inhibitory complex with Swi6[HP1] within a heterochromatin restricted genomic context.

The question of whether Epe1 harbors any histone demethylase activity remains unanswered. Inspired by earlier studies on the *Drosophila* dKDM4A protein where HP1 binding stimulates demethylase activity, we attempted to reconstitute the enzymatic function of Epe1 in the presence of Swi6[HP1] (*Lin et al., 2008*). However, our studies are unable to detect Epe1 mediated enzymatic demethylation in the presence or absence of Swi6[HP1] (*Figure 2—figure supplement 2A*). It is noteworthy that a histone demethylase like protein in *Neurospora*, DMM-1 shares many similarities with Epe1 and also surprisingly lacks any in vitro enzymatic activity (*Honda et al., 2010*). DMM-1 also interacts with the HP1 homolog in *Neurospora* leading us to speculate that this surprising mode of heterochromatin regulation we uncovered in the case of Epe1 might also extend to other fungal systems. Given that expressing the C-terminus of Epe1 only partially restores wild-type levels of Epe1 activity (*Figure 3C,D*), we speculate that there could potentially be unique substrates or conditions required to reconstitute the enzymatic activity of Epe1. We favor a model in which heterochromatin associated proteins could act as positive or negative allosteric regulators of Epe1. For example, Bdf2, a BRD4 homolog that localizes to heterochromatin boundaries along with Epe1, is a possible candidate that may activate the latent enzymatic functions of Epe1 as H3K9 methylation levels are negligible at the heterochromatin-euchromatin boundary (*Wang et al., 2013*).

Our observations add to an expanding list of proteins that mimic histone modifiers but have built-in non-enzymatic functions that regulate the establishment and maintenance of epigenetic states. The *Drosophila* protein dKDM4A is a prominent example of a histone demethylase with enzymatic and non-enzymatic functions (*Colmenares et al., 2017*). More extreme examples of histone-modifying enzyme mimicry are observed in the case of JARID2, a subunit of the PRC2 complex that shares several characteristics with JmjC domain-containing proteins but has a structural role in regulating PRC2 complex assembly (*Kasinath et al., 2018*; *Son et al., 2013*). Proteins involved in the *Arabidopsis* RNA-dependent DNA methylation pathway, SUVH2 and SUVH9, resemble SET domain methyltransferases but lack enzymatic activity. Instead, both proteins recognize methylated DNA and are involved in RNA pol V recruitment in plants to establish epigenetic silencing (*Johnson et al., 2008*; *Law et al., 2013*). Although Epe1 has a JmjC domain, it is possible that its putative catalytic function has been repurposed to regulate its interaction with Swi6[HP1]. Our data suggest that H3K9 methylation promotes the cooperative assembly of complexes between Epe1 and Swi6[HP1] at sites of heterochromatin formation.

We speculate that the enzymatic and non-enzymatic functions of Epe1 are likely to oppose heterochromatin assembly on different timescales. As a first approximation, the stabilization of Epe1 at sites of heterochromatin formation is sufficient to displace histone deacetylases and hasten the loss of transcriptional silencing. Hence, the non-enzymatic function of Epe1 serves as the first line of opposition to heterochromatin establishment. This transient inhibition of heterochromatin assembly is likely to be followed by a slower enzymatic step where Epe1 might demethylate H3K9 methylated histones or activate transcription (*Sorida et al., 2019*). Our work provides a biochemical basis for how chromatin associated factors that are thought to purely function as enzymes have non-enzymatic properties that regulate heterochromatin assembly and epigenetic inheritance.

## Materials and methods

### Key resources table

| Reagent type (species) or resource | Designation | Source or reference | Identifiers | Additional information |
|---|---|---|---|---|
| Chemical compound, drug | Trizma base | Sigma aldrich | Cat# T1503-5KG | |
| Chemical compound, drug | Boric acid | Sigma Aldrich | Cat# B6768-5KG | |
| Chemical compound, drug | EDTA | Sigma Aldrich | Cat# EDS-500G | |

*Continued on next page*

Continued

| Reagent type (species) or resource | Designation | Source or reference | Identifiers | Additional information |
|---|---|---|---|---|
| Chemical compound, drug | DTT | Thermo Fisher Scientific | Cat# BP17225 | |
| Chemical compound, drug | HEPES | Sigma Aldrich | Cat# H4034-100G | |
| Chemical compound, drug | Proteinase K | Thermo Fisher Scientific | Cat# FEREO0491 | |
| Chemical compound, drug | Magnesium chloride hexahydrate | Sigma Aldrich | Cat# M2670-500G | |
| Chemical compound, drug | Potassium chloride | Sigma Aldrich | Cat# P9541-500G | |
| Chemical compound, drug | Sodium chloride | Thermo Fisher Scientific | Cat# S271-3 | |
| Chemical compound, drug | Sodium acetate | Sigma Aldrich | Cat# S2889-250G | |
| Chemical compound, drug | Tryptone | Thermo Fisher Scientific | Cat# BP1421-500 | |
| Chemical compound, drug | SDS micropellets | Thermo Fisher Scientific | Cat# BP8200-500 | |
| Chemical compound, drug | Agarose | Thermo Fisher Scientific | Cat# BP1356-500 | |
| Chemical compound, drug | PMSF | Calbiochem | Cat# 7110–5 GM | |
| Chemical compound, drug | Triton X-100 | Sigma Aldrich | Cat# T8787-250ML | |
| Chemical compound, drug | Lithium chloride | Sigma Aldrich | Cat# L4408-500G | |
| Chemical compound, drug | Lithium acetate dihydrate | Sigma Aldrich | Cat# CAS6108-17-4 | |
| Chemical compound, drug | Tween 20 | Thermo Fisher Scientific | Cat# CAS9005-64-5 | |
| Chemical compound, drug | Peptone | RPI Research Products | Cat# P20240-1000.0 | |
| Chemical compound, drug | Leupeptin | RPI Research Products | Cat# L22035-0.025 | |
| Chemical compound, drug | Aprotinin | RPI Research Products | Cat# A20550-0.05 | |
| Chemical compound, drug | Pepstatin | RPI Research Products | Cat# P30100-0.025 | |
| Chemical compound, drug | N-Oxalylglycine | Sigma Aldrich | Cat# O9390-10MG | |
| Chemical compound, drug | α-Ketoglutaric acid disodium salt dihydrate | Sigma Aldrich | Cat# 75892–25G | |
| Chemical compound, drug | SuperSignal West Pico PLUS Chemiluminescent Substrate | Thermo Scientific | Cat# B2162617 | |
| Chemical compound, drug | Trichloroacetic acid | Sigma Aldrich | Cat# T0699-100ML | |
| Chemical compound, drug | Phenol:chloroform: isoamyl alcohol | Sigma Aldrich | Cat# P3803-100ML | |
| Chemical compound, drug | L-ascorbic acid | Fisher Chemical | Cat# C6H8O6 | |

*Continued*

| Reagent type (species) or resource | Designation | Source or reference | Identifiers | Additional information |
|---|---|---|---|---|
| Chemical compound, drug | Coblat(II) chloride hexahydrate | Sigma Aldrich | Cat# 255599–100G | |
| Chemical compound, drug | Ammonium iron (II) sulfate hexahydrate | ACROS ORGANICS | Cat# 423721000 | |
| Chemical compound, drug | Formaldehyde | Sigma Aldrich | Cat# 252549–500 ML | |
| Chemical compound, drug | Acetonitrile | OmniSolv | Cat# AX0156-1 | |
| Chemical compound, drug | Trifluoroacetic acid | Fisher Chemical | Cat# A116—10 × 1 AMP | |
| Chemical compound, drug | Dimethyl pimelimidate | Thermo Scientific | Cat# 21667 | |
| Chemical compound, drug | Ammonium hydroxide solution | Sigma Aldrich | Cat# 221-228-100ML-A | |
| Chemical compound, drug | Ethanolamine | Sigma Aldrich | Cat# 411000–100 ML | |
| Chemical compound, drug | Ethyleneglycol bis succinimidylsuccinate | Thermo Scientific | Cat# 21565 | |
| Chemical compound, drug | Glycogen | Sigma Aldrich | Cat# 10901393001 | |
| Chemical compound, drug | GelGreen Nucleic Acid Stain | BioTium | Cat# 41004 | |
| Chemical compound, drug | 30% Acrylamide/ Bis solution, 37.5:1 | Bio-Rad | Cat# 1610158 | |
| Chemical compound, drug | Ammonium persulfate | Bio-Rad | Cat# A3678-100G | |
| Chemical compound, drug | TEMED | Sigma Aldrich | Cat# T9281-50ml | |
| Chemical compound, drug | Hydrochloric acid | Thermo Fisher Scientific | Cat# A14-500 | |
| Chemical compound, drug | Agar | Sigma Aldrich | Cat# A1296-1KG | |
| Chemical compound, drug | IPTG | Thermo Fisher Scientific | Cat# BP1755-10 | |
| Antibody | Anti-H3K9me2 (Mouse monoclonal) | Abcam | Cat# ab1220 RRID:AB_449854 | IF (1:1000), WB (1:2500) |
| Antibody | Anti-H3K9me3 (Rabbit polyclonal) | Abcam | Cat# ab8898 RRID:AB_306848 | IF (1:1000), WB (1:5000) |
| Antibody | Anti-H3 (Rabbit polyclonal) | Abcam | Cat# ab1791 RRID:AB_302613 | IF (1:1000), WB (1:5000) |
| Antibody | Anti-H3K4me3 (Rabbit polyclonal) | Abcam | Cat# ab8580 RRID:AB_2827504 | IF (1:1000), WB (1:1000) |
| Antibody | Monoclonal ANTI-FLAG M2 Antibody | Abcam | Cat# F1804-5MG RRID:AB_262044 | IF (1:5000), WB (1:5000) |
| Antibody | THE V5 Tag Antibody | GenScript | Cat# A01724-100 RRID:AB_2827501 | IF (1:1000), WB (1:5000) |
| Antibody | Anti-MBP (Mouse monoclonal) | New England Biolabs | Cat# E8032S RRID:AB_2827502 | WB (1:5000) |
| Antibody | Anti-Swi6 (Rabbit polyclonal) | Custom | | WB (1:2500) |
| Peptide, recombinant protein | H3K9me3 (1603 Da) | New England Peptide | Custom | H2N-ARTKQTAR(K9me3) STGGKA-amide |

*Continued on next page*

*Continued*

| Reagent type (species) or resource | Designation | Source or reference | Identifiers | Additional information |
|---|---|---|---|---|
| Peptide, recombinant protein | H3K9me0 (1560 Da) | New England Peptide | Custom | ARTKQTKARKSTGGKA-amide |
| Peptide, recombinant protein | H3K9(me2) (2751 Da) | Anaspec peptide | Cat# AS-64359 | H-ARTKQTARK(ME2)STGGKA PPKQLAGGK(biotin)-OH |
| Peptide, recombinant protein | H3K9(me3) (2766 Da) | Anaspec peptide | Cat# AS-64360 | H-ARTKQTARK(ME3)STGGK APPKQLAGGK(biotin)-OH |
| Peptide, recombinant protein | H3 (2722 Da) | Anaspec peptide | Cat# AS-61702 | H-ARTKQTARKSTGGKAP PKQLAGGK(biotin)-OH |
| Other | Dynabeads Protein G | Thermo Fisher Scientific | LOT# 00448217 | |
| Other | Dynabeads M-280 Streptavidin | Thermo Fisher Scientific | LOT# 00448388 | |
| Other | Dynabeads Protein A | Thermo Fisher Scientific | LOT# 00689576 | |
| Other | ANTI-FLAG M2 Affinity Gel | Sigma Aldrich | LOT# A2220-5ML | |
| Other | Amylose Resin | New England Biolabs | LOT# E8021L | |
| Other | Pierce Gutathione Agarose, 100 ml | Thermo Scientific | LOT# R1241698 | |

## Plasmids

Plasmids containing Epe1 wild-type and point mutants were constructed by modifying existing pFA6a C-terminal tagging plasmids. Point mutations were introduced by designing primers using guidelines described in Quick Change mutagenesis protocols. A ligation independent cloning approach was used to construct pFastBac vectors containing wild-type Epe1 and Epe1 H297A mutant for recombinant protein expression and also for other MBP fusion constructs for *E. coli* expression. 3X FLAG Swi6[HP1] and 3X FLAG Chp2[HP1] were cloned into existing pGEX vectors downstream of the Prescission protease cleavage site using Gibson assembly. The construction of the *10X gal4-10X tetO-ade6*[+] plasmid involved modifying plasmids containing a 10X tetO sequence and subcloning Gal4 UAS sequences derived from a *Drosophila* pVALIUM 10X UAS vector. Vectors containing Gal4-Clr3 or Gal4-Sir2 were made using a modified pDual vector with an nmt1 promoter that enables facile integration of DNA sequences at the *leu1* locus in fission yeast (*Matsuyama et al., 2004*). Further details regarding plasmid construction are readily available upon request.

## Strains

All strains were constructed using a PCR-based gene targeting approach (*Bähler et al., 1998*). In cases where we generated point mutations of *epe1*, we reintroduced the full length wild-type or mutant gene in *epe1Δ* strains. All strains were genotyped using colony PCR assays. We subsequently verified protein expression using western blots for each of the mutant strains. Strains with *10X gal4-10X tetO-ade6*[+] were constructed using a 5-Fluoroorotic Acid (FOA) selection strategy based on disrupting the endogenous *ura4* locus. Strains with Gal4-Clr3 or Gal4-Sir2 were made by digesting pDual vectors with a Not1 restriction enzyme followed by transformations and -LEU based selection. Other deletions of heterochromatin associated factors were achieved either by PCR-based gene targeting approaches or by a cross followed by random spore analysis and PCR based screening to select for colonies that harbored the reporter gene. All strains used in this study are listed in *Supplementary file 1*, Table S1. Further details regarding strain construction are available upon request.

## Cell lysis, co-immunoprecipitation and western blotting

1.5 L of fission yeast cells cells were grown in YEA medium at 32°C to an $OD_{600}$ = 3.5 and harvested by centrifugation. The cell pellets were washed with 10 ml TBS pH 7.5, re-suspended in 1.5 ml lysis buffer (30 mM HEPES pH 7.5, 100 mM NaCl, 0.25% Triton X-100, 5 mM $MgCl_2$, 1 mM DTT), and

the cell suspension was snap-frozen into liquid nitrogen to form yeast 'balls' and cryogenically ground using a SPEX 6875D Freezer/Mill. The frozen cell powder was thawed at room temperature and re-suspended in an additional 10 ml of lysis buffer with protease inhibitor cocktail and 1 mM PMSF. The cell lysates were subjected to two rounds of centrifugation at 18000 rpm for 5 and 30 mins in a JA-25.50 rotor (Beckman). Bradford assay was used to normalize protein levels for co-immunoprecipitation and immunoblot analysis.

Protein G Magnetic Beads were pre-incubated with antibody for 4 h and crosslinked with 10 volumes of crosslinking buffer containing 20 mM DMP (3 mg DMP/ml of 0.2 M Boric Acid pH 9) for 30 min at room temperature by rotating. Crosslinking was quenched by washing twice and incubated with 0.2 M ethanolamine pH 8 for 2 h at room temperature by rotating. The cell lysates were then incubated with antibody crosslinked beads for 3 h at 4°C. Beads were washed three times in 1 ml lysis buffer for 5 mins each, then eluted with 500 μl of 10 mM ammonium hydroxide. The ammonium hydroxide was evaporated using speed vac (SPC-100H) for 5 h and re-suspended in SDS sample buffer. Samples were resolved on SDS–polyacrylamide gel electrophoresis (SDS-PAGE) and transferred to PVDF membranes. Immunoblotting was performed by blocking PVDF membrane in Tris-buffered saline (TBS) pH 7.5 with 0.1% Tween-20 (TBST) containing 5% non-fat dry milk and subsequently probed with desired primary antibodies and secondary antibodies. Blots were developed by enhanced chemiluminescence (ECL) method and detected with Bio-Rad ChemiDoc Imaging System. All co-IP experiments were reproduced N = 2.

## Chromatin immunoprecipitation (ChIP)

Cells were grown till late log phase (OD 600–1.3 to 1.8) in yeast extract supplemented with adenine (YEA) or YEA containing tetracycline (2.5 μg/ml) medium and fixed with 1% formaldehyde for 15 min at room temperature (RT). 130 mM glycine was then added to quench the reaction and incubated for 5 min at RT. The cells were harvested by centrifugation, and washed twice with TBS (50 mM Tris, pH 7.6, 500 mM NaCl). Cell pellets were resuspended in 300 μl lysis buffer (50 mM HEPES-KOH, pH 7.5, 100 mM NaCl, 1 mM EDTA, 1% Triton X-100, 0.1% SDS, and protease inhibitors) to which 500 μl 0.5 mm glass beads were added and cell lysis was carried out by bead beating using Omni Bead Ruptor at 3000 rpm × 30 s × 10 cycles. Tubes were punctured and the flow-through was collected in a new tube by centrifugation which was subjected to sonication to obtain fragment sizes of roughly 100–500 bp long. After sonication the extract was centrifuged for 15 min at 13000 rpm at 4°C. The soluble chromatin was then transferred to a fresh tube and normalized for protein concentration by the Bradford assay. For each normalized sample, 25 μl lysate was saved as input, to which 225 μl of 1xTE/1% SDS were added (TE: 50 mM Tris pH 8.0, 1 mM EDTA). Dynabeads Protein A were preincubated with Anti-H3K9me2 antibody (PRID:AB_449854). For each immunoprecipitation, 2 μg antibody coupled to 30 μl beads was added to 400 μl soluble chromatin, and the final volume of 500 μl was achieved by adding lysis buffer. Samples were incubated for 2 h at 4°C, the beads were collected on magnetic stands, and washed three times with 1 ml lysis buffer and once with 1 ml TE. For eluting bound chromatin, 100 μL elution buffer I (50 mM Tris pH 8.0, 10 mM EDTA, 1% SDS) was added and the samples were incubated at 65°C for 5 min. The eluate was collected and incubated with 150 μl 1xTE/0.67% SDS in the same way. Input and immunoprecipitated samples were finally incubated overnight at 65°C to reverse crosslink for more than 6 h. 60 μg glycogen, 100 μg proteinase K (Roche), 44 μl of 5M LiCl, and 250 μl of 1xTE was added to each sample and incubation was continued at 55°C for 1 h. Phenol/chloroform extraction was carried out for all the samples followed by ethanol precipitation. Immuno-precipitated DNA was resuspended in 100 μl of 10 mM Tris pH 7.5 and 50 mM NaCl and was used for qPCR (SYBR Green) using an Eppendorf Mastercycler Realplex. For extra crosslinking, prior to fixing with 1% formaldehyde, the cultures were incubated at 18°C for 2 h in a shaking incubator. The cells were pelleted and resuspended in 4.5 ml of 1x PBS. To this 1.5 mM EGS (ethylene glycol bis[succinimidylsuccinate]), Pierce (Fisher) was added and the samples were incubated at RT for 20 min with mild shaking before adding 1% formaldehyde. The samples were then processed as mentioned above. All ChIP experiments were reproduced N = 2.

## Recombinant protein purification from insect cells and *E. coli*

MBP-His-TEV-Epe1 and Epe1-ΔC were cloned into a pFastBac vector (Thermo Fisher Scientific) and used for Bacmid generation. Low-titer baculoviruses were produced by transfecting Bacmid into

Sf21 cells using Cellfectin II reagent (Gibco). Full-length *S. pombe* Epe1 protein (wild-type and mutant) was expressed in Hi5 cells infected by high titer baculovirus which was amplified from Sf21 cells. After 44 h of infection, Hi5 cells were harvested and lysed in buffer A (30 mM Tris-HCl (pH 8.0), 500 mM NaCl, 5 mM EDTA, 5 mM β-mercaptoethanol with protease inhibitor cocktails) using Emulsiflex-C3 (Avestin). The cleared cell lysate was applied to Amylose resin (New England Biolabs) followed by washing with buffer A and elution with buffer A containing 10 mM maltose. The N-terminal His-MBP tag can be removed by TEV protease cleavage, which was used to evaluate protein solubility. Proteins were further purified using a Superdex 200 (GE Healthcare) size exclusion column. The protein was concentrated in a storage buffer containing 30 mM Tris-HCl (pH 8.0), 500 mM NaCl, 30% glycerol, and 1 mM TCEP.

Proteins were expressed in BL21 (DE3) cells. Cells were grown to log phase at 37°C, cooled on ice, and induced with 0.3 mM IPTG before incubation for 18 h at 18°C. Pellets were suspended in tris buffered saline (TBS) and frozen at −80°C until further use. For purification, cell pellets were thawed in lysis buffer (500 mM NaCl, 50 mM Tris pH 7.5, 10% glycerol) supplemented with protease inhibitor and cells were ruptured using sonicator. Cell debris was removed by centrifugation and the supernatant was incubated with appropriate beads for each purification for 3 h at 4°C. We used a GST tag and glutathione beads (GST) beads for 3X FLAG Swi6$^{HP1}$, Swi6$^{HP1}$ W104A, Swi6$^{HP1}$ loop-X mutant and 3X FLAG-Chp2$^{HP1}$ purifications. Swi6$^{HP1}$ and Chp2$^{HP1}$ were subject to overnight cleavage with Prescission protease. We used an MBP tag and amylose resin for the purification of Epe1$^{434-948}$ and Epe1$^{434-600}$. After washing, Epe1$^{434-948}$ and Epe1$^{434-600}$ were eluted with elution buffer (lysis buffer + 20 mM maltose + 5 mM EDTA). To purify Swi6$^{HP1}$ used for in vitro binding assays, we used a hexahistidine tag and Nickel resin. After elution, the N-terminal 6X His tag was removed using a SUMO protease followed by addition purification using an anion exchange column.

## In vitro binding assay

In vitro binding assays were performed by immobilizing recombinant 3X FLAG- Swi6$^{HP1}$ or 3X FLAG-Chp2$^{HP1}$ on 25 µl of FLAG M2 beads, which were incubated with three different concentrations of recombinant MBP fusion proteins in 600 µl binding buffer containing 20 mM HEPES pH 7.5, 150 mM NaCl, 5 mM MgCl$_2$, 10% glycerol, 0.25% Triton -X 100, 1 mM DTT. Reactions were incubated at 4°C for 2 h and washed three times in 1 ml washing buffer (20 mM HEPES pH 7.5, 150 mM NaCl, 5 mM MgCl$_2$, 10% glycerol, 0.25% Triton -X 100, 1 mM DTT) for 5 min each, then 30µl of SDS sample buffer was added followed by incubation at 95°C for 5 min. Proteins were separated through SDS-PAGE and transferred to PVDF membrane followed by incubation with anti-MBP monoclonal antibody (E8032S, NEB) and M2 Flag antibody (A8592, Sigma). Depending on the experiment, we added co-factors 100µM ammonium iron (II) sulfate hexahydrate and 1 mM α-ketoglutarate or 5µg of H3 peptides (1–21 amino acids) with or without modifications. Western blot data for in vitro binding assays were analyzed using ImageJ software. The exposure times for the interaction assays were chosen and differ in each experiment to capture differences in the interaction between Epe1 and Swi6 depending on the assay conditions. Assays performed on different blots cannot be compared but samples loaded on the same blot can be readily compared to each other. All in vitro binding experiments were reproduced N ≥ 3.

## Demethylase assay

Mass spectrometry-based demethylase assays were performed using 5µg MBP-Epe1, 10µg Swi6, and 20µM peptide (either H3K9me3 or H3K9me2). The peptide sequences used in these assays were as follows: 1) NH$_2$-ARTKQTAR(K9me3)STGGKA-amide (H3K9me3, 1–15 amino acids). 2) H-ARTKQTARK(K9me2)STGGKAPRKQLA - OH) (H3K9me2, 1–21 amino acids). The demethylase assay reaction buffer consists of 50 mM HEPES (pH 7.5), 50 mM NaCl, 100 µM ammonium iron (II) sulfate hexahydrate, 1 mM L-ascorbic acid, and 1 mM α-ketoglutarate. Reaction mixtures were incubated at 37°C for 3 h, quenched with an equal volume of 1% trifluoroacetic acid, and stored at −20°C. In parallel, we also performed demethylase assays using equivalent amounts of purified JMJD2A (protein amounts equalized using SDS-PAGE gels). Samples were thawed and desalted using a ZipTip (Millipore). The ZipTip was first equilibrated twice with wetting solution (50% acetonitrile) and twice with equilibration solution (0.1% trifluoroacetic acid). Samples (10µl) following the demethylase assay were washed with washing solution (0.1% trifluoroacetic acid) before elution with 4µl of 0.1%

trifluoroacetic acid/50% acetonitrile. Matrix-assisted laser desorption ionization (MALDI) mass spectrometry was performed using a Waters Tofspec-2E in reflectron mode with delayed extraction (Department of Chemistry, University of Michigan). All demethylase experiments were reproduced N = 2.

## In vitro translation (IVT) assays

To identify minimal Epe1 fragments that bind to Swi6$^{HP1}$, Epe1 fragments were translated in vitro using TNT T7-coupled reticulocyte lysate (Promega) with $^{35}$S-labeled methionine (Roche). In vitro translated target proteins were incubated with Flag-tagged Swi6 at 4°C for 20 min. M2 FLAG beads pre-equilibrated with buffer B containing 30 mM Tris-HCl (pH 8.0), 50 mM NaCl, 1 mM DTT, and 0.1% NP-40 (w/v) were mixed and incubated at 4°C for 45 min with rotation. The beads were washed three times with buffer B, and bead-bound proteins were separated by SDS-PAGE. Dried gels were analyzed by overnight exposure of a phosphor imager plate.

## In vitro binding assays using fission yeast cell extracts

To generate fission yeast cell lysates, we grew 100 ml of fission yeast cells in YEA medium at 32°C to an OD$_{600}$ = 3–3.5 and harvested cells by centrifugation. The cell pellets were washed with 1 ml TBS pH 7.5 and resuspended in lysis buffer (30 mM HEPES pH 7.5, 100 mM NaCl, 0.25% Triton X-100, 5 mM MgCl2, 1 mM DTT). 0.5 mm glass beads were added and cell lysis was carried out by bead beating using Omni Bead Ruptor at 3000 rpm (30 s x eight cycles). The cell extract was centrifuged for 20 min at 15000 rpm at 4°C and the lysates were incubated with beads pre-bound with recombinant MBP-Epe1$^{434-948}$ protein for 3 h at 4°C. Beads were washed three times with 1 ml lysis buffer and proteins were eluted by boiling the beads in SDS sample buffer. Proteins were resolved by SDS-PAGE and analyzed by immunoblotting with appropriate primary and secondary antibodies.

To test whether the N- and C-terminal fragments of Epe1 binds in *trans*, we used the following protocol: 1.5 l of fission yeast cells expressing 3XFLAG- Epe1-N (1-434) were grown in YEA medium at 32°C to an OD$_{600}$ = 3.5 and harvested by centrifugation. The cell pellets were washed with 10 ml TBS pH 7.5, re-suspended in 1.5 ml lysis buffer (30 mM HEPES pH 7.5, 100 mM NaCl, 0.25% Triton X-100, 5 mM MgCl$_2$,1 mM DTT), and the cell suspension was snap-frozen into liquid nitrogen to form yeast 'balls' and cryogenically ground using a SPEX 6875D Freezer/Mill. The frozen cell powder was thawed at room temperature and re-suspended in an additional 10 ml of lysis buffer (30 mM HEPES pH 7.5, 100 mM NaCl, 0.25% Triton X-100, 5 mM MgCl$_2$, 1 mM DTT) with protease inhibitor cocktail and 1 mM PMSF. The cell lysates were subjected to two rounds of centrifugation at 18000 rpm in a JA-25.50 rotor (Beckman). Protein G Magnetic Beads were pre-incubated with an M2 FLAG antibody and the antibodies were crosslinked to the beads prior to usage. The cell lysates expressing 3X FLAG-Epe1-N (1-434) were then incubated with the M2 FLAG antibody conjugated beads overnight at 4°C. Beads were washed once in 1 ml lysis buffer. Importantly, 3XFLAG-Epe1-N (1-434) was retained on beads for subsequent binding assays. Recombinant Swi6$^{HP1}$ (1 μg) was pre-incubated with recombinant MBP-Epe1$^{434-948}$ for 1 h in lysis buffer, then this was added to beads pre-bound with the 3XFLAG-Epe1-N (1-434) fragment.

In assays where we tested the effect of modified peptides on the *trans* interaction between the N- and C-terminal halves of Epe1, we added 1 μg of either H3K9me0 or K9-trimethyl H3 (H3K9me3) peptides. Immunoblotting was performed by blocking PVDF membrane in TBS pH 7.5 with 0.1% Tween-20 (TBST) containing 5% non-fat dry milk and subsequently probed with desired primary antibodies and secondary antibodies. Blots were developed with the ECL method and detected with Bio-Rad ChemiDoc Imaging System.

## Streptavidin pull-down assay

H3K9me0 or H3K9me3 biotinylated peptides (50 nM) were pre-incubated with either recombinant MBP-Epe1 or 3XFLAG-Swi6 in binding buffer (30 mM Tris-HCl [pH 7.5], 600 mM NaCl, 1% Triton X-100, 5% glycerol, 2.5% BSA) for 1 h at 4°C. Then streptavidin M280 beads (Invitrogen) were added to the pre-mixed protein-peptide mixture and incubated for an additional 2 h at 4°C. The beads were then rinsed three times with wash buffer (30 mM Tris-HCl [pH 7.5], 600 mM NaCl, 1% Triton X-100, 5% glycerol, 2.5% BSA) and bound proteins were eluted by boiling the beads in SDS sample

buffer. The input and bound proteins were resolved by SDS-PAGE and analyzed by immunoblotting with MBP antibody (E8032S, NEB).

MBP pull-down assays were performed using Calf thymus histones (Sigma) to evaluate Epe1 binding specificity. 1 µg of Calf thymus histones (Sigma) were pre-incubated with recombinant MBP-Epe1 in binding buffer (30 mM Tris-HCI [pH 7.5], 600 mM NaCl, 1% Triton X-100, 5% glycerol, 2.5% BSA) for 1 h at 4°C. Amylose resin was added to the binding assay and incubated for an additional 2 h at 4°C. Beads were rinsed three times with wash buffer (30 mM Tris-HCI [pH 7.5], 600 mM NaCl, 1% Triton X-100, 5% glycerol, 2.5% BSA) and bound proteins were eluted by boiling the beads in SDS sample buffer. The input and bound proteins were resolved by SDS-PAGE and analyzed by immunoblotting with MBP antibody (E8032S, NEB).

## Acknowledgements

We thank Patrick O'Brien and the Chromatin Club at the University of Michigan for insightful suggestions and critical feedback. We thank Ryan Baldridge for his detailed critique of the manuscript, coffee, and advice for experiments. We thank Gayathri Santanu for her exceptional support for GR. We thank Hiten Madhani, Danesh Moazed, and Songtao Jia for yeast strains. We thank Geeta Narlikar for sharing the Chp2 expression plasmid. We also thank Danesh Moazed for sharing the Swi6 primary antibody, which was used in all co-immunoprecipitation and chromatin immunoprecipitation (ChIP) experiments. This work was supported by startup funds from the Regents of the University of Michigan.

## Additional information

### Funding

| Funder | Grant reference number | Author |
|---|---|---|
| National Institute of General Medical Sciences | Michigan Predoctoral Training in Genetics T32GM007544 | Ajay Larkin |
| UM Rogel Cancer Center | Research Grant | Sojin An |
| University of Michigan | Startup funds | Gulzhan Raiymbek Nidhi Khurana Saarang Gopinath Saikat Biswas Kaushik Ragunathan |
| National Institute of Diabetes and Digestive and Kidney Diseases | DK111465 | Uhn-soo Cho |
| National Science Foundation | CHE-1508492 | Raymond C Trievel |

Thefundershad no role in study design, data collection and interpretation, or the decision to submit the work for publication.

### Author contributions

Gulzhan Raiymbek, Conceptualization, Investigation, Methodology; Sojin An, Uhn-soo Cho, Resources, Investigation, Methodology; Nidhi Khurana, Formal analysis, Validation, Investigation; Saarang Gopinath, Saikat Biswas, Investigation; Ajay Larkin, Formal analysis, Investigation; Raymond C Trievel, Resources; Kaushik Ragunathan, Conceptualization, Resources, Formal analysis, Supervision, Funding acquisition, Methodology, Project administration

### Author ORCIDs

Ajay Larkin https://orcid.org/0000-0002-5967-9535
Uhn-soo Cho http://orcid.org/0000-0002-6992-2455
Kaushik Ragunathan https://orcid.org/0000-0003-4776-8589

Decision letter and Author response
Decision letter https://doi.org/10.7554/eLife.53155.sa1
Author response https://doi.org/10.7554/eLife.53155.sa2

## Additional files

### Supplementary files
- Supplementary file 1. Description of genotypes of strains used in this study.
- Supplementary file 2. Description of plasmids used in this study.
- Transparent reporting form

### Data availability
All data generated or analysed during this study are included in the manuscript and supporting files.

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
