## [Decision Letter]

**Acceptance summary:**

This paper identifies a non-enzymatic function in counteracting heterochromatin for the putative histone demethylase Epe1. The presented data suggest that binding of Epe1 to histone H3 methylated at lysine 9 – a hallmark of heterochromatin – enables Epe1 to interact with the key heterochromatin protein Swi6 by releasing an autoinhibitory conformation. The results highlight the possibility that putative enzymes may have non-catalytic modes of action that are as important or more important than catalytic ones.

**Decision letter after peer review:**

[Editors’ note: the authors submitted for reconsideration following the decision after peer review. What follows is the decision letter after the first round of review.]

Thank you for submitting your work entitled "An auto-inhibitory mechanism regulates the non-enzymatic functions of a histone demethylase" for consideration by *eLife*. Your article has been reviewed by four peer reviewers, and the evaluation has been overseen by a Reviewing Editor and a Senior Editor. The following individual involved in review of your submission has agreed to reveal their identity: Yota Murakami (Reviewer #3).

Our decision has been reached after consultation between the reviewers. Based on these discussions and the individual reviews below, we regret to inform you that your work in its present form will not be considered further for publication in *eLife*.

The reviewers agreed that the mechanism by which Epe1 counteracts heterochromatic silencing is of great interest. The reviewers were also intrigued by the proposal that Epe1 autoinhibition is relieved by interaction with H3K9me, which enables Epe1 to bind Swi6, thereby displacing heterochromatin-promoting factors such as Clr3. However, there was strong consensus that key aspects of the model – Epe1 autoinhibition, and the release of autoinhibition by direct interaction with H3K9me – are not strongly supported by the data. The proposed Epe1 noncatalytic mechanism was also not strongly supported and the overall model of Epe1 action has important unstated implication that should be considered and discussed. We would be willing to consider a resubmitted manuscript that convincingly demonstrates how H3K9me affects Epe1 conformation and function and presents a clear, well-supported model of Epe1 anti-silencing activity.

Reviewer #1:

This manuscript investigates how Epe1, a putative histone demethylase for which catalytic activity has not been demonstrated, counteracts epigenetic silencing in *S. pombe*. The authors show that mutations in residues predicted to be involved in cofactor binding disrupt Epe1 activity, prevent its localization to heterochromatin, and disrupt its interaction with Swi6. Loss of Swi6 or H3K9me prevents Epe1 from localizing to heterochromatin. The authors show that Epe1 interacts with Swi6 via a region adjacent to the JmjC domain, that this domain interacts more strongly with Swi6 in isolation than in the context of the entire Epe1 protein, and that expression of this domain (lacking JmjC) can counteract heterochromatic silencing. The authors also show that the N- and C-terminal halves of Epe1 interact in vitro. Interaction between Epe1 and Swi6 is stimulated by H3K9me in vitro, and a mutation that reduces Swi6 affinity for H3K9me does not eliminate this effect. Finally, the authors show that tethering the histone deacetylase Clr3 to heterochromatin can overcome the anti-silencing function of Epe1. The authors propose that Epe1 exists in an auto-inhibited conformation which is relieved by interaction with H3K9me, allowing Epe1 to interact with Swi6, which in turn counteracts silencing via a non-enzymatic mechanism that involves displacing silencing proteins, such as Clr3, that are recruited by Swi6.

This manuscript contains a great deal of interesting results, but I feel that two of the key conclusions are supported by weak or ambiguous data.

First, the authors propose that Epe1 has a noncatalytic mode of anti-silencing. This is based on (1) the ability of a Swi6-interactin domain (lacking JmjC) of Epe1 to counteract heterochromatic silencing, (2) the ability of cofactor binding mutations to disrupt Epe1 localization, and (3) the demonstration that tethering the histone deacetylase Clr3 to heterochromatin can overcome the anti-silencing function of Epe1. However, the first result is quite artificial and does not show that the whole Epe1 protein, expressed at endogenous levels, has a noncatalytic anti-silencing mode. The authors don't know how their mutations interfere with Swi6 binding, and therefore the second result doesn't to me argue for a noncatalytic activity. Same of Clr3 tethering – I find it equally plausible that Clr3 and Epe1 have antagonistic catalytic activities, and tethering Clr3 tips the balance in favour of heterochromatin. I don't find the claim for a noncatalytic function of Epe1 to be strongly supported.

Second, the authors propose that binding of H3K9me to Epe1 relieves Epe1 autoinhibition and enables Epe1 to interact with Swi6. This is supported by the observations that (1) a domain of Epe1 interacts more strongly than the entire protein with Swi6 in vitro, that (2) the N- and C-terminal halves of Epe1 interact in vitro, that (3) the interaction between Epe1 and Swi6 is stimulated by H3K9me in vitro, that (4) a mutation that reduces Swi6 affinity for H3K9me does not eliminate this effect, and that (5) several mutations disrupt Epe1 localization to heterochromatin and binding to Swi6. However, the first two lines of evidence do not demonstrate autoinhibition, they just show in vitro effects that are consistent with autoinhibition and may not be relevant in vivo. The evidence that H3K9me stimulates the interaction with Swi6 via binding to Epe1 is weak and indirect. The authors do not demonstrate H3K9me binding to Epe1 and do not identify the domain responsible for this. The authors also do not know how the cofactor binding mutations disrupt Epe1 interaction with Swi6. The conclusion that binding of H3K9me to Epe1 relieves Epe1 autoinhibition and enables Epe1 to interact with Swi6 remains a hypothesis.

I am particularly concerned by the weak evidence because I find the overall model proposed by the authors counterintuitive (or at least nonintuitive). A simple model would be that the interaction between H3K9me and Swi6 increases the local concentration of Swi6 at heterochromatin (facilitated by the protein's phase transition properties), which allows Epe1 to bind. But the authors add a major complication – interaction between H3K9me and Epe1 is also required. In this model, Epe1 must interact with H3K9me and Swi6, but it is also competing with Swi6 for H3K9me binding. Is this competition important? Can tethering of Swi6 to chromatin recruit Epe1 in the absence of H3K9me? I feel that the authors' model needs substantially more support than is currently provided.

Reviewer #2:

In fission yeast, Epe1 is a putative histone demethylase that plays important roles in regulating the heterochromatin landscape. However, the mechanism by which Epe1 functions is not clear.

In this manuscript, the authors found that point mutations that abolish co-factor binding in the JmjC domain can disrupt Epe1's binding with Swi6 and also reduces Epe1 localization at heterochromatin. in vitro binding assays confirmed that one Epe1 mutant (H297A) has significantly reduced binding with Swi6. Truncation analysis mapped the Swi6 interaction interface to Epe1 433-630, which is outside of the JmjC domain. Moreover, Epe1 433-630 interacts with Swi6 stronger than full length Epe1, which led the authors to hypothesize that full-length Epe1 might adopt an auto-inhibitive conformation to prevent its binding with Swi6. Indeed, the authors found that Epe1's N-terminal region can bind to its C-terminal region. Authors further demonstrated that H3K9me3 is required for Swi6-Epe1 interaction in vivo and that H3K9me3 peptide enhanced Swi6-Epe1 interaction in vitro, but not Epe1-H29&A. These results suggest a mechanism for Epe1 to only interact with Swi6 at heterochromatin regions. Lastly, the authors showed that Epe1 blocks Swi6's interaction with Clr3, an HDAC important for heterochromatin assembly.

Overall, the data quality is high and the proposed model is very interesting. The findings will be of high interest to the heterochromatin field. I would recommend its publication after addressing several major questions.

1) The model suggests that the JmjC domain of Epe1 binds to H3K9me3 and that the H297A mutation affects this binding. Since the authors have recombinant proteins at hand, they should test these interactions.

2) The model also suggests that in the presence of H3K9me3, interaction between Epe1's N- and C-terminal regions will be reduced. This should also be tested.

3) In Figure 4, there appears to be very weak binding of WT Epe1 with Swi6 without H3K9me3 peptide. This is very different from results shown in Figures 2 and 3. What's the reason for this discrepancy?

4) Clr3∆ or Sir2∆ has different effects on H3K9me at centromeres, mating type region, and telomeres. If the hypothesis is correct, it is expected that Epe1 localization will change differently at different locations. ChIP analysis of Epe1 will complement Figure 4B and strengthen the hypothesis.

*Reviewer #3:*

The putative histone demethylase Epe1 in fission yeast has been shown to localize heterochromatin via interaction with a HP1 homologue Swi6 and antagonize heterochromatin formation probably through its JmjC-domain dependent demethylation activity, though the enzymatic activity has not been detected in vitro. In this manuscript, authors made several interesting observations. (1) JmjC-domain mutants of Epe1 loose interaction with Swi6 and heterochromatin localization. (2) Epe1 interacts with Swi6 via its C-terminal half that does not include JmjC domain. (3) Epe1 N-terminal half including JmjC-domain interacts with C-terminal half in *trans*. (5) H3K9me peptide enhance interaction between Swi6 and Epe1. (4) Expression of C-terminal half Epe1 antagonize maintenance of heterochromatic silencing with non-enzymatic activity. (6) Mutation of JmjC domain strengthen the interaction between Swi6 and Clr3, a Swi6-binding silencing factor. (7) Artificial recruitment of Clr3 supports maintenance of heterochromatin in the presence of wild type Epe1. Taken together authors claimed that Epe1-Swi6 binding is auto-inhibited by cis-interaction between N-terminal half and C-terminal half of Epe1. The auto-inhibition can be released by H3K9 methylation and Epe1 compete with Clr3 for Swi6-binding resulting in removal of heterochromatin.

Author's model is novel and interesting in terms of regulatory mechanism of heterochromatin by Epe1. However, their model is not fully supported by their experimental data. There is no direct evidence showing Epe1 N-terminal half and C-terminal half interacts in cis. Their result can be also explained by *trans*-interaction. Moreover, they did not present direct evidence showing this interaction is inhibitory to Swi6 binding. This could be tested by adding N-terminal half in the in vitro binding assay of Epe1^434-948^ and Swi6 (Figure 3A). In addition, If H3K9me peptide release the auto inhibition through Epe1 as author claimed, addition of H3K9me peptide in the in vitro binding assay would disrupt interaction between N- and C-terminal half of Epe1. Furthermore, I think one of the main point of their model is that competition between Epe1 and Clr3 for Swi6, by which Epe1 inhibits stable maintenance of artificially formed heterochromatin. However, the evidence for this mechanism is not enough. At least, they should check localization of Clr3 at artificial heterochromatin by ChIP analysis. If their model is correct, amount of Clr3 at artificial heterochromatin (-Tet) would increase in *epe1*∆ or *epe1*-H297A cells and decreased in the presence of Epe1 or Epe1^434-948^.

Recent paper (Sorida et al., 2019), showed that N-terminal domain that does not include jmjC domain has transcription activation activity and deletion of this domain caused severe loss of Swi6 binding and heterochromatin localization of Epe1. In addition, they showed H297A mutation also caused reduction of Swi6 binding. Probably, this paper published after the submission of this manuscript, but authors should mention and discuss about this paper in the revised manuscript.

In conclusion, this manuscript presents novel data and their model will attract interests of many readers, but the points I described above should be clarified before publication.

[Editors’ note: further revisions were suggested prior to acceptance, as described below.]

Thank you for submitting your article "An auto-inhibitory mechanism regulates the non-enzymatic functions of a histone demethylase" for consideration by *eLife*. Your article has been reviewed by four peer reviewers, one of whom is a member of our Board of Reviewing Editors, and the evaluation has been overseen by Kevin Struhl as the Senior Editor. The following individuals involved in review of your submission have agreed to reveal their identity: Songtao Jia (Reviewer #2); Yota Murakami (Reviewer #4).

The reviewers have discussed the reviews with one another and the Reviewing Editor has drafted this decision to help you prepare a revised submission.

Summary:

Since the discovery of the JmjC domain as the catalytic region of histone demethylases, the molecular function of Epe1 in fission yeast has been under intense debate, and no demethylase activity has been shown. Interest in this protein intensified with the finding that H3K9me can be inherited without initiation signals, but only in an *epe1* mutant background. The assumption has been that Epe1 is a H3K9 demethylase, with the supporting evidence that *epe1* mutants expected to abolish demethylase activity also abolished Epe1 function. This manuscript provides detailed characterization of these mutations, challenging the hypothesis that they abolish demethylase activity. Instead, this paper identifies a non-enzymatic function for Epe1 – binding to Swi6 – that is regulated by H3K9me. The authors' data also suggest, but do not prove yet, that binding of Epe1 to H3K9me enables Epe1 to interact with Swi6 by releasing an autoinhibitory conformation. The results highlight the possibility that proteins that appear to be primarily enzymes may have non-catalytic modes of action that are as important or more important than catalytic ones.

Essential revisions:

1) The most important outstanding concern is about the interaction between H3K9me3 and Epe1. This interaction is central to the authors' hypothesis yet is tested only with the entire Epe1 protein and reported in two supplementary panels. One concern is that the model presented by the authors – that Swi6 and Epe1 may not compete for H3K9me tails – depends on the binding affinities of these proteins for H3K9me. There are significantly fewer H3K9me tails (~2000 nucleosomes) in fission yeast than Swi6 molecules (~20,000, Sadaie, 2008), which would argue that only molecules at the affinity of Swi6 for H3K9me2/3, or tighter, could access tails. Co-occupancy of Swi6 and Epe1 of adjacent tails therefore is possible only in a circumstance where the affinity of Epe1 is at least as high as that of Swi6 (which is not terribly high, measured at ~2-15 µM depending on the group), given the substantial Swi6 excess. The easiest way to show that H3K9me binding by Epe1 directs its assembly on chromatin is to mutate the region or residues required for this recognition and test its impact.

Given the many available truncation constructs, the authors should be able to identify the domain that interacts with H3K9me and determine the affinity of the interaction. As previously suggested, the JmjC domain is a good candidate for H3K9me binding. These data could be used to generate a mutant that abolishes binding, and the mutant phenotypes should be tested in vivo. These experiments would be very important to support the authors' conclusions.

2) The presented data clearly indicate that an H3K9me peptide can disrupt the interaction between the N-terminal and C-terminal domains of Epe1. However, the authors have not tested if the addition of the N-terminal half inhibits the interaction between the C-terminus and Swi6. This would provide support for the authors' autoinhibition model.

3) The presented experiments strongly suggest, but do not prove, an autoinhibitory mechanism, as there are no experiments that directly address Epe1 conformation. Therefore, claims regarding such a mechanism should be softened. The title could be changed to "Interaction with histone H3 methylated at lysine 9 regulates the non-enzymatic functions of the putative histone demethylase Epe1" or something similar, and strong claims like "An auto-inhibitory conformation regulates the non-enzymatic properties of Epe1 through its interaction with Swi6" in the Abstract should be amended or removed.

---

## [Author Response]

[Editors’ note: the authors resubmitted a revised version of the paper for consideration. What follows is the authors’ response to the first round of review.]

The reviewers agreed that the mechanism by which Epe1 counteracts heterochromatic silencing is of great interest. The reviewers were also intrigued by the proposal that Epe1 autoinhibition is relieved by interaction with H3K9me, which enables Epe1 to bind Swi6, thereby displacing heterochromatin-promoting factors such as Clr3.However, there was strong consensus that key aspects of the model – Epe1 autoinhibition, and the release of autoinhibition by direct interaction with H3K9me – are not strongly supported by the data. The proposed Epe1 noncatalytic mechanism was also not strongly supported and the overall model of Epe1 action has important unstated implication that should be considered and discussed. We would be willing to consider a resubmitted manuscript that convincingly demonstrates how H3K9me affects Epe1 conformation and function and presents a clear, well-supported model of Epe1 anti-silencing activity.

Since our initial submission, we have added experimental data that addresses some of the key issues highlighted by reviewers and editors, including, support for a model based on auto-inhibition, H3K9 methylation dependent release from auto-inhibition and additional evidence for a non-catalytic function associated with Epe1. We have also edited our Discussion section in light of recent work from the Murakami lab outlining a non-enzymatic function for the Epe1 N-terminus (Sorida et al., 2019). We have considered the implications of our work and how this affects current models of epigenetic inheritance in a revised Discussion section. The result of our efforts is twelve new figure panels, substantial edits to the manuscript and a critical assessment of reviewer feedback.

This significant update to the manuscript was made possible by four key experiments (along with several minor/additional revisions) that we have summarized here:

1) We detected a direct binding interaction between Epe1 and an H3K9me3 peptide and H3K9 methylated histones. Epe1 also exhibits specificity towards H3K9 methylated as opposed to H3K4 methylated histones (Figure 2—figure supplement 1H-I).

2) We also directly demonstrate that the interaction between the Epe1 N- and C-terminus in *trans* can be disrupted upon addition of an H3K9me3 peptide. These results lend strong support to a model where Epe1 is preserved in an auto-inhibited state with H3K9 methylation reversing this inhibition (new Figure 5A).

3) We identified and purified a C-terminal truncation mutant of Epe1, (Epe1-ΔC) which exhibits a substantial increase in its interaction with Swi6 relative to the full-length protein even in the absence of a modified peptide. Furthermore, Epe1-ΔC does not undergo an H3K9me3 dependent enhancement in its interaction with Swi6 unlike its full-length counterpart.

Collectively, these results clearly support our notion that the C-terminus of Epe1 plays a critical role in enforcing an H3K9 methylation dependent mode of interaction with Swi6 (new Figure 5B-E).

We designed a completely novel ectopic silencing approach in fission yeast to provide additional evidence for a non-catalytic function associated with the Epe1 C-terminus. Since Epe1 interacts with Swi6 and excludes other effector molecules, we hypothesized that Swi6 mediated heterochromatin initiation would be highly sensitive to Epe1 activity (enzymatic or nonenzymatic). We expressed the C-terminus of Epe1 (434-948) which lacks the JmjC domain in cells where a TetR-Swi6 CSD fusion has been tethered. The C-terminal fragment expression now completely blocks heterochromatin formation at the ectopic site. These results clearly support a model where the non-enzymatic function of Epe1 relies on its Swi6 interaction (Figure 3E and Figure 3C).

Reviewer #1:[…] This manuscript contains a great deal of interesting results, but I feel that two of the key conclusions are supported by weak or ambiguous data.

We thank the reviewer for his/her positive comments about our findings. We have sought to address concerns raised about the ambiguity of our data and our interpretations through additional experiments. Our new experiments lend strong support to our proposed model and we hope that our new findings will alleviate these concerns.

First, the authors propose that Epe1 has a noncatalytic mode of anti-silencing.

The premise of our studies is based on prior work revealing a non-enzymatic basis for Epe1 in resetting epigenetic silencing (Trewick et al., 2005; Zofall et al., 2006). The overexpression of Epe1 point mutants that lack any catalytic functions completely restores normal levels of reporter gene silencing in *epe1*Δ cells. Recent studies have further substantiated these early findings and suggest that Epe1 might have a critical role in activating transcription within pericentromeric repeats (Bao et al., 2019; Sorida et al., 2019). Our studies, viewed in this broader context, adds a strong mechanistic perspective to this urgent question through stringent biochemical analysis.

This is based on (1) the ability of a Swi6-interactin domain (lacking JmjC) of Epe1 to counteract heterochromatic silencing, (2) the ability of cofactor binding mutations to disrupt Epe1 localization, and (3) the demonstration that tethering the histone deacetylase Clr3 to heterochromatin can overcome the anti-silencing function of Epe1. However, the first result is quite artificial and does not show that the whole Epe1 protein, expressed at endogenous levels, has a noncatalytic anti-silencing mode. The authors don't know how their mutations interfere with Swi6 binding, and therefore the second result doesn't to me argue for a noncatalytic activity.

We thank the reviewer for this important critique. Although we do not know the mechanism that leads to the loss of interaction between Epe1 JmjC mutants and Swi6, we can unambiguously catalog the molecular consequence of the loss of their interaction. We demonstrate that a loss of interaction between the full-length Epe1 JmjC mutant (expressed at endogenous levels) and Swi6 is compensated by a gain of an interaction between a heterochromatin enhancer, Clr3 and Swi6 (Figure 6A). Therefore, we disagree that our manuscript does not establish a non-enzymatic function associated with Epe1 when the full-length protein has been expressed at endogenous levels.

Our tethering experiments were designed based on findings in Figure 6A to overcome the need for Swi6 to be involved in Clr3 recruitment to sites of heterochromatin formation. We were surprised that the tethering experiments conformed to our expectations where Clr3 recruitment appears to be rate-limiting for epigenetic inheritance. Intriguingly, our ectopic system recapitulates a DNA sequence dependent maintenance mechanism where Atf1/Pcr1 (two DNA binding proteins) are thought to recruit Clr3 to the fission yeast mating type locus (Wang et al., 2017). Therefore, the Clr3 tethering approach recapitulates what potentially occurs in fission yeast cells at natural loci.

Our Discussion section is agnostic about the possibility of Epe1 having an additional enzymatic function. We certainly have not ruled out this possibility. We have stated that the expression of the Epe1 C-terminus only partially suppresses heterochromatin assembly. It is eminently possible that our reconstitution conditions failed to capture histone demethylation. Our Discussion section highlights some potential strategies to consider while attempting to reconstitute its putative catalytic functions. We have added the following lines to the Discussion:

“Nevertheless, given the similarity of the JmjC domain of Epe1 to active demethylases, we favor a model where different proteins that interact with Epe1 could act as positive or negative allosteric regulators. For example, Bdf2, a BRD4 homolog that localizes to heterochromatin boundaries along with Epe1, represents a possible candidate that activates its latent enzymatic functions given that H3K9 methylation levels are negligible at the heterochromatin-euchromatin interface (Wang et al., 2013).”

Same of Clr3 tethering – I find it equally plausible that Clr3 and Epe1 have antagonistic catalytic activities, and tethering Clr3 tips the balance in favour of heterochromatin. I don't find the claim for a noncatalytic function of Epe1 to be strongly supported.

It is important to note that whether Clr3 acts directly or indirectly to oppose Epe1 activity does not negate the central claims of our manuscript. Nevertheless, we deeply appreciate and recognize the need to be cautious with regards to our conclusions. Hence, we have amended the Results section of the manuscript relating to the Clr3 tethering experiments to explicitly state:

“We hypothesized that directly tethering Clr3 at an ectopic site would eliminate its dependency on Swi6 to localize at sites of heterochromatin formation. In this genetic context, we rationalized that Epe1 would be unlikely to counteract Clr3 recruitment. One caveat of this experimental strategy is that it does not rule out the possibility that Epe1 and Clr3 have antagonistic catalytic activities but simply seeks to test whether Clr3 recruitment itself might be rate-limiting at sites of heterochromatin formation.”

Second, the authors propose that binding of H3K9me to Epe1 relieves Epe1 autoinhibition and enables Epe1 to interact with Swi6. This is supported by the observations that (1) a domain of Epe1 interacts more strongly than the entire protein with Swi6 in vitro, that (2) the N- and C-terminal halves of Epe1 interact in vitro, that (3) the interaction between Epe1 and Swi6 is stimulated by H3K9me in vitro, that (4) a mutation that reduces Swi6 affinity for H3K9me does not eliminate this effect, and that (5) several mutations disrupt Epe1 localization to heterochromatin and binding to Swi6. However, the first two lines of evidence do not demonstrate autoinhibition, they just show in vitro effects that are consistent with autoinhibition and may not be relevant in vivo. The evidence that H3K9me stimulates the interaction with Swi6 via binding to Epe1 is weak and indirect. The authors do not demonstrate H3K9me binding to Epe1 and do not identify the domain responsible for this. The authors also do not know how the cofactor binding mutations disrupt Epe1 interaction with Swi6. The conclusion that binding of H3K9me to Epe1 relieves Epe1 autoinhibition and enables Epe1 to interact with Swi6 remains a hypothesis.

We have now included three independent lines of experimental evidence to support our claims:

1) We tested the direct interaction between Epe1 and an H3K9me3 peptide and H3K9 methylated histones. Our binding assays clearly detect a direct interaction between Epe1 and an H3K9me3 peptide as opposed to an unmodified H3K9me0 peptide. Furthermore, Epe1 selectively interacts with H3K9 methylated histones as opposed to H3K4 methylated histones. These data have now been added to the revised version of the manuscript (Figure 2—figure supplement 1H and I).

2) To identify a potential domain within Epe1 that enforces an H3K9 methylation dependent mode of interaction with Swi6, we generated a C-terminal truncation mutant (Epe1-ΔC). Using in vitroand in vivomeasurements: We demonstrate that Epe1-ΔC exhibits a significantly higher degree of binding to Swi6 (Figure 5B-C). Furthermore, Epe1-ΔC fails to exhibit any H3K9me3 peptide dependent stimulation (Figure 5D). Epe1-ΔC binds to Swi6 in cells in the presence or absence of the H3K9 methyltransferase, *clr4*D (Figure 5E). We conclude that the C-terminus of Epe1 enforces an H3K9 methylation dependent mode of interaction between Epe1 and Swi6.

These data have now been added as a new figure to the revised version of our manuscript (new Figure 5B-E).

3) To test whether H3K9 methylation reverses an auto-inhibitory conformation of Epe1, we measured the interaction between the N- and C-terminal Epe1 fragments in the presence or absence of an H3K9me3 peptide. The addition of an H3K9me3 peptide disrupts the direct interaction between the N- and C-terminal Epe1 fragments in *trans.* This data has now been added to the revised version of the manuscript (new Figure 5A).

I am particularly concerned by the weak evidence because I find the overall model proposed by the authors counterintuitive (or at least nonintuitive). A simple model would be that the interaction between H3K9me and Swi6 increases the local concentration of Swi6 at heterochromatin (facilitated by the protein's phase transition properties), which allows Epe1 to bind. But the authors add a major complication – interaction between H3K9me and Epe1 is also required.

We initially considered and spent considerable time and effort exploring a model where Swi6 phase separation facilitates Epe1 binding. However, new and existing biochemical lines of evidence that we have presented in this manuscript forced us to re-evaluate these early assumptions. In particular, the most striking findings that contradict this model are:

1) A Swi6 chromodomain mutant (Swi6 W104A) that fails to bind to H3K9 methylated histones nevertheless undergoes a substantial increase in its interaction with Epe1 in the presence of an H3K9me3 peptide (Figure 4—figure supplement 1D).

2) An Epe1 C-terminal truncation mutant, Epe1-ΔC, does not undergo any stimulation in its interaction with wild-type Swi6 in the presence of an H3K9me3 peptide (Figure 5D).

3) A *trans* interaction between the Epe1 N- and C- terminus can be disrupted by an H3K9me3 peptide bolstering our case for a model predicated on Epe1 being responsive to the presence of H3K9 methylation (Figure 5A).

Based on these observations, we envision a model where H3K9 methylation promotes the cooperative assembly of a heterochromatin inhibitory complex comprised of Epe1 and Swi6. The model where the phase separation properties of Swi6 drive Epe1 recruitment is not fully supported by our data.

In this model, Epe1 must interact with H3K9me and Swi6, but it is also competing with Swi6 for H3K9me binding. Is this competition important?

The question of competing interactions between Epe1 and Swi6 for H3K9 methylated histone tails is undoubtedly an interesting and contentious point to consider. This is never an issue in our experiments where our in vitroassays are supplemented with an excess of histone H3K9me3 peptide. In cells, it is equally plausible that the multivalency of heterochromatin leads to non-competitive complex assembly wherein Epe1 and Swi6 bind to adjacent or distal nucleosomes. Given the dense networks of H3K9 methylation at sites of constitutive heterochromatin, we favor a model where the two proteins need not have to compete for the same binding site. We have added a line in our Discussion section which highlights how Swi6 and Epe1 could potentially compete for a shared binding site:

“The ability of Epe1 to bind to H3K9 methylation raises the possibility that it competes with Swi6 for a shared binding site. However, heterochromatin consists of dense networks of H3K9 methylation. It is possible that Epe1 while interacting with Swi6 could undergo stimulation via an interaction with an adjacent or a distal nucleosome eliminating a need for competition.”

Can tethering of Swi6 to chromatin recruit Epe1 in the absence of H3K9me? I feel that the authors' model needs substantially more support than is currently provided.

We interpreted this request by the reviewer as having to show further evidence for the non-enzymatic properties of Epe1 by developing new phenotypic assays. We were inspired by this question to design the experiment detailed in this response.

Two hybrid experiments where Epe1 is fused to an activation domain and Swi6 is fused to a DNA binding domain detect a direct interaction between the two proteins (Sorida et al., 2019; Braun et al., 2011). The two hybrid assay also detects a loss of interaction between Epe1 JmjC mutants and Swi6 (Sorida et al., 2019). Therefore, tethering Swi6 to chromatin can recruit Epe1 to activate reporter gene expression in budding yeast cells which lack H3K9 methylation. However, these findings are not at all in conflict with our data. We clearly demonstrate the existence of a weak interaction between Epe1 and Swi6 and then go on to show that this interaction is enhanced in the presence of H3K9 methylation (Figure 2 and Figure 5). Since our reconstitution experiments are performed in the absence of any additional cellular factors, we believe that our measurements a unique feature of Epe1 that cannot be detected using genetic assays.

Furthermore, we designed a completely novel ectopic silencing approach in fission yeast to provide additional evidence for a non-catalytic function associated with the Epe1 C-terminus. Since Epe1 interacts with Swi6 and excludes other effector molecules, we hypothesized that Swi6 mediated heterochromatin initiation would be highly sensitive to Epe1 activity (enzymatic or non-enzymatic). Tethering Swi6 at an ectopic site does not lead to heterochromatin establishment in fission yeast. Deleting *epe1*D was sufficient to trigger heterochromatin assembly and mediate epigenetic inheritance where silencing is maintained in the absence of the initiator. Next, we expressed the C-terminus of Epe1 (434-948) which lacks the JmjC domain in cells where a TetR-Swi6 CSD fusion has been tethered. The C-terminal fragment expression now completely blocks heterochromatin formation at the ectopic site. These results clearly support a model where the non-enzymatic function of Epe1 relies on its Swi6 interaction. Our in vitroand in vivodata seek to demonstrate how modified chromatin domains provide a context where this interaction is significantly strengthened.

Reviewer #2:In fission yeast, Epe1 is a putative histone demethylase that plays important roles in regulating the heterochromatin landscape. However, the mechanism by which Epe1 functions is not clear.In this manuscript, the authors found that point mutations that abolish co-factor binding in the JmjC domain can disrupt Epe1's binding with Swi6 and also reduces Epe1 localization at heterochromatin. in vitro binding assays confirmed that one Epe1 mutant (H297A) has significantly reduced binding with Swi6. Truncation analysis mapped the Swi6 interaction interface to Epe1 433-630, which is outside of the JmjC domain. Moreover, Epe1 433-630 interacts with Swi6 stronger than full length Epe1, which led the authors to hypothesize that full-length Epe1 might adopt an auto-inhibitive conformation to prevent its binding with Swi6. Indeed, the authors found that Epe1's N-terminal region can bind to its C-terminal region. Authors further demonstrated that H3K9me3 is required for Swi6-Epe1 interaction in vivo and that H3K9me3 peptide enhanced Swi6-Epe1 interaction in vitro, but not Epe1-H29&A. These results suggest a mechanism for Epe1 to only interact with Swi6 at heterochromatin regions. Lastly, the authors showed that Epe1 blocks Swi6's interaction with Clr3, an HDAC important for heterochromatin assembly.Overall, the data quality is high and the proposed model is very interesting. The findings will be of high interest to the heterochromatin field. I would recommend its publication after addressing several major questions.1) The model suggests that the JmjC domain of Epe1 binds to H3K9me3 and that the H297A mutation affects this binding. Since the authors have recombinant proteins at hand, they should test these interactions.

We tested the direct interaction between Epe1 and an H3K9me3 peptide and H3K9 methylated histones. Our binding assays clearly detect a direct interaction between Epe1 and an H3K9me3 peptide as opposed to an unmodified H3K9me0 peptide. Furthermore, Epe1 selectively interacts with H3K9 methylated histones as opposed to H3K4 methylated histones. These data have now been added to the revised version of the manuscript (Figure 2—figure supplement 1H and I).

2) The model also suggests that in the presence of H3K9me3, interaction between Epe1's N- and C-terminal regions will be reduced. This should also be tested.

To test whether H3K9 methylation releases an auto-inhibitory conformation of Epe1, we measured the interaction between the N- and C-terminal Epe1 fragments in the presence or absence of an H3K9me3 peptide. The addition of an H3K9me3 peptide disrupts the direct interaction between the N- and C-terminal Epe1 fragments in *trans.* This data has now been added to the revised version of the manuscript (Figure 5A).

3) In Figure 4, there appears to be very weak binding of WT Epe1 with Swi6 without H3K9me3 peptide. This is very different from results shown in Figures 2 and 3. What's the reason for this discrepancy?

We apologize to the reviewer for the lack of clarity in our presentation. This difference is due to the use of chemiluminescent based western blot assays to measure binding efficiencies. We have explicitly stated that these assays are semi-quantitative in the manuscript text although we do obtain trends based on adding different Epe1 protein concentrations. Although samples in the same blot can be compared, we cannot compare samples loaded on different gels.

In Figure 2E where we compare the interaction between wild-type Epe1 and Epe1 H297A we use high exposure times to reveal a difference in binding. To demonstrate a stimulation in the interaction between Epe1 and Swi6 in the presence of an H3K9me3 peptide, we used a low exposure time so that the assay is sensitive to reveal a substantial change in binding upon peptide addition. In this assay as well, the difference between wild-type Epe1 and Epe1 H297A binding to Swi6 is still visible but less prominent (Figure 4E).

We have now included an additional statement in our Materials and methods section regarding how we choose exposure times to capture differences in binding interactions. The Materials and methods section describing how we setup our binding assays has been amended as follows:

“The exposure times for the interaction assays were chosen and differ in each experiment in order to capture differences in the interaction between Epe1 and Swi6 depending on the assay conditions. Assays performed on different blots cannot be compared but samples loaded on the same blot can be readily compared to each other”.

4) Clr3∆ or Sir2∆ has different effects on H3K9me at centromeres, mating type region, and telomeres. If the hypothesis is correct, it is expected that Epe1 localization will change differently at different locations. ChIP analysis of Epe1 will complement Figure 4B and strengthen the hypothesis.

We are very grateful to the reviewer for this suggestion. We deleted *clr3∆* in strains expressing Epe1-3XFLAG. We cross-linked cells with formaldehyde and performed ChIP experiments to map Epe1 localization at the pericentromeric repeats, mating-type locus and the telomeres. The *dg* repeats exhibit a 2-3 fold increase in Epe1 localization in *clr3*Δcells (Shimada et al.). This increase is consistent with a model where Clr3 and Epe1 trade places at sites of heterochromatin formation. Epe1 localization also exhibits a two-fold increase at the mating type locus in strains where the *cenH* nucleation site is intact. Deleting *clr3*Δ causes a complete loss of Epe1 localization at the telomeres consistent with a critical role for the SHREC complex in telomere silencing (Sugiyama et al., 2007).

Reviewer #3:The putative histone demethylase Epe1 in fission yeast has been shown to localize heterochromatin via interaction with a HP1 homologue Swi6 and antagonize heterochromatin formation probably through its JmjC-domain dependent demethylation activity, though the enzymatic activity has not been detected in vitro. In this manuscript, authors made several interesting observations. (1) JmjC-domain mutants of Epe1 loose interaction with Swi6 and heterochromatin localization. (2) Epe1 interacts with Swi6 via its C-terminal half that does not include JmjC domain. (3) Epe1 N-terminal half including JmjC-domain interacts with C-terminal half in trans. (5) H3K9me peptide enhance interaction between Swi6 and Epe1. (4) Expression of C-terminal half Epe1 antagonize maintenance of heterochromatic silencing with non-enzymatic activity. (6) Mutation of JmjC domain strengthen the interaction between Swi6 and Clr3, a Swi6-binding silencing factor. (7) Artificial recruitment of Clr3 supports maintenance of heterochromatin in the presence of wild type Epe1. Taken together authors claimed that Epe1-Swi6 binding is auto-inhibited by cis-interaction between N-terminal half and C-terminal half of Epe1. The auto-inhibition can be released by H3K9 methylation and Epe1 compete with Clr3 for Swi6-binding resulting in removal of heterochromatin.

We are very grateful to Dr. Murakami for his encouraging and positive comments regarding our manuscript. Furthermore, we were not aware of a publication from his lab (Sorida et al., 2019) relating to the non-enzymatic functions of Epe1 at the time of submission. We have corrected this oversight on our part. We have also highlighted areas of overlap and how both our studies collectively add important mechanistic insight relating to the non-enzymatic functions of Epe1 (albeit using different phenotypic assays). We have attempted to address all of the major revisions outlined in Dr. Murakami’s scholarly critique.

Author's model is novel and interesting in terms of regulatory mechanism of heterochromatin by Epe1. However, their model is not fully supported by their experimental data. There is no direct evidence showing Epe1 N-terminal half and C-terminal half interacts in cis. Their result can be also explained by trans-interaction. Moreover, they did not present direct evidence showing this interaction is inhibitory to Swi6 binding. This could be tested by adding N-terminal half in the in vitro binding assay of Epe1^434-948^ and Swi6 (Figure 3A). In addition, If H3K9me peptide release the auto inhibition through Epe1 as author claimed, addition of H3K9me peptide in the in vitro binding assay would disrupt interaction between N- and C-terminal half of Epe1.

We thank Dr. Murakami for his positive feedback about our manuscript. We have added additional experiments that lends strong support to our model. To test whether H3K9 methylation releases an auto-inhibitory conformation of Epe1, we measured the interaction between the N- and C-terminal Epe1 fragments in the presence or absence of an H3K9me3 peptide. First, we purified Epe1 (1-434) from fission yeast cells using 3XFLAG epitope tag. Despite protein expression being low in fission yeast cells, we obtained sufficient quantities of the N-terminal fragment in order to setup in vitrobinding assays. Next, we added a recombinant Epe1 (434-948) protein fragment purified from *E. coli*. We performed experiments in the presence and absence of an H3K9me3 peptide. The addition of an H3K9me3 peptide disrupts a direct interaction between the N- and C-terminal Epe1 fragments in *trans.* This data has now been added to the revised version of the manuscript (new Figure 5A). This data strongly supports our hypothesis where an interaction between the Epe1 N- and C-terminal halves is dependent on H3K9 methylation.

Furthermore, I think one of the main point of their model is that competition between Epe1 and Clr3 for Swi6, by which Epe1 inhibits stable maintenance of artificially formed heterochromatin. However, the evidence for this mechanism is not enough. At least, they should check localization of Clr3 at artificial heterochromatin by ChIP analysis. If their model is correct, amount of Clr3 at artificial heterochromatin (-Tet) would increase in epe1∆ or epe1-H297A cells and decreased in the presence of Epe1 or Epe1^434-948^.

Our conclusions in this part of the manuscript builds on previous studies revealing an interplay between Epe1 and Clr3 in localizing at sites of heterochromatin formation (Shimada et al., 2009). Inspired by this study, we tested whether Clr3 could interact with Swi6 and how this interaction is altered in an Epe1 JmjC mutant background (Figure 6A).

To test whether Clr3 binding is critical to preserve epigenetic memory, we fused a 3X-V5 epitope tag to the Clr3 C-terminus and performed ChIP experiments in cells expressing WT Epe1 or an Epe1 H297A mutant. The V5 epitope tag was chosen since it is orthogonal to other FLAG tagged constructs that are expressed in this strain background. We performed ChIP experiments using formaldehyde or after adding reactive cross-linkers in addition to formaldehyde. Under both conditions, we were unable to detect Clr3 enrichment at the ectopic site relative to controls where Clr4 is absent. Furthermore, we note that this problem was not unique to the ectopic locus since we also could not detect significant Clr3 enrichment at the *dg* repeats compared to *clr4*Δcontrol strains. We compared our studies to those performed by other groups including Dr.Murakami’s lab (Shimada et al., 2010). One important difference is the use a 3X FLAG epitope tag versus a 3X V5 epitope tag.

However, we would like to emphasize that our ectopic system is explicitly dependent on Clr3 for heterochromatin establishment. The deletion of *clr3*Δleads to a complete loss of epigenetic silencing where cells turn white in the absence of tetracycline (Ragunathan et al., 2015). The deletion of Epe1 (*epe1*D *clr3*Δ)cannot rescue this loss of silencing. Therefore any circumstance where we see maintenance mandates the presence of Clr3. We fully expect that our ChIP experiments will conform to the proposed model where Epe1 and Clr3 trade places while interacting with Swi6 to preserve epigenetic memory. Additional experiments requested by reviewer 2 (new Figure 6C-E) also support this hypothesis.

Recent paper (Sorida et al., 2019), showed that N-terminal domain that does not include jmjC domain has transcription activation activity and deletion of this domain caused severe loss of Swi6 binding and heterochromatin localization of Epe1. In addition, they showed H297A mutation also caused reduction of Swi6 binding. Probably, this paper published after the submission of this manuscript, but authors should mention and discuss about this paper in the revised manuscript.

We apologize to Dr. Murakami for this oversight. We were not aware of the publication of this manuscript at the time of our submission. In combination with Bao et al., 2019; we believe that the publication of our manuscript will be a timely addition to demonstrate how a broad spectrum of non-enzymatic functions associated with Epe1 affect the establishment and maintenance of epigenetic silencing in fission yeast cells.

[Editors’ note: what follows is the authors’ response to the second round of review.]

Essential revisions:1) The most important outstanding concern is about the interaction between H3K9me3 and Epe1. This interaction is central to the authors' hypothesis yet is tested only with the entire Epe1 protein and reported in two supplementary panels. One concern is that the model presented by the authors – that Swi6 and Epe1 may not compete for H3K9me tails – depends on the binding affinities of these proteins for H3K9me. There are significantly fewer H3K9me tails (~2000 nucleosomes) in fission yeast than Swi6 molecules (~20,000, Sadaie, 2008), which would argue that only molecules at the affinity of Swi6 for H3K9me2/3, or tighter, could access tails. Co-occupancy of Swi6 and Epe1 of adjacent tails therefore is possible only in a circumstance where the affinity of Epe1 is at least as high as that of Swi6 (which is not terribly high, measured at ~2-15 µM depending on the group), given the substantial Swi6 excess. The easiest way to show that H3K9me binding by Epe1 directs its assembly on chromatin is to mutate the region or residues required for this recognition and test its impact.Given the many available truncation constructs, the authors should be able to identify the domain that interacts with H3K9me and determine the affinity of the interaction. As previously suggested, the JmjC domain is a good candidate for H3K9me binding. These data could be used to generate a mutant that abolishes binding, and the mutant phenotypes should be tested in vivo. These experiments would be very important to support the authors' conclusions.

We thank the reviewers for their scholarly comments and for highlighting a major point of interest in our manuscript. We have generated new supporting data to highlight the unique H3K9me specific binding properties associated with Epe1. We have also elevated the importance of these findings by relocating the peptide binding and histone binding assay to the main figure panel (new Figure 5).

The previous version of our manuscript established that the full-length Epe1 protein preferentially interacts with an H3K9me3 peptide (Figure 5A) and specifically binds to H3K9me3 histones (as opposed to an H3K4me3 histone, Figure 5B). Now, we have used a C-terminal truncation mutant of Epe1 (Epe1-ΔC) to test whether the H3K9me3 binding activity resides within the N-terminus of the protein. Indeed, the Epe1-ΔC protein (deletion of the Epe1 C-terminus, 600-948 amino acids) exhibits a preferential interaction with an H3K9me3 peptide (new Figure 5—figure supplement 1A) and specifically binds to H3K9me3 histones (new Figure 5—figure supplement 1B). Since the N-terminal fragment of Epe1 includes the putative catalytic JmjC domain, we are fully in agreement with reviewers that this domain might be directly responsible for H3K9me binding.

To purify The JmjC domain of Epe1 and design mutants to test H3K9 methylation binding, we dedicated considerable effort to test different solubility and affinity tags in *E. coli* for recombinant protein expression. However, despite our best efforts we were unable to purify sufficient amounts of recombinant Epe1 protein required for rigorous affinity measurements using either ITC or fluorescence polarization. Therefore, we are unable to compare and report the absolute H3K9 methylation binding affinities associated with Epe1 and Swi6.

However, we fully recognize the importance of this question raised by the reviewers and endeavored to provide further support for our proposed model. Swi6 is preserved in an auto-inhibited state through interactions between adjacent chromodomains (Canzio et al., 2013). A Swi6 Loop-X mutation (Swi6 R93A K94A) releases the chromodomain from an auto-inhibited conformation enabling increased engagement with an H3K9me3 peptide. Despite the lack of auto-inhibition, the interaction between Epe1 and a Swi6 Loop-X mutant is also stimulated by the presence of an H3K9me3 peptide (new Figure 4—figure supplement 1E). Therefore, Epe1 remains responsive to the presence of an H3K9 methylated peptide even in an experimental context where Swi6 is constitutively released from auto-inhibition.

The central premise of our model is that H3K9 methylation acts via Epe1 to stimulate complex formation with Swi6. Our multi-pronged genetic and biochemical analyses provide strong evidence in favor of this model. We believe that the lack of affinity measurements does not detract from the major conclusions in our revised submission.

2) The presented data clearly indicate that an H3K9me peptide can disrupt the interaction between the N-terminal and C-terminal domains of Epe1. However, the authors have not tested if the addition of the N-terminal half inhibits the interaction between the C-terminus and Swi6. This would provide support for the authors' autoinhibition model.

We thank the reviewers for suggesting a key experiment that would lend additional support for our model of H3K9 methylation dependent regulation of Epe1 activity. Our previous studies demonstrated that the N-terminal fragment of Epe1 (1-434 amino acids) interacts with a C-terminal fragment (434-948 amino acids) in *trans.* We also observed that the addition of an H3K9me3 peptide but not an H3K9me0 peptide disrupts a *trans* interaction between the two protein fragments (Figure 5C). We purified limited quantities of the Epe1-N (1-434) fragment using an M2 FLAG antibody. The protein was retained on beads and immediately used for subsequent binding assays. We added recombinant MBP-Epe1-C (434-948) in the presence of 1.5-2 fold molar excess of Swi6 (or control experiments where Swi6 was left out). Our in vitro binding experiments demonstrate that Swi6 could also efficiently disrupt complex formation between the Epe1 N-terminus and C-terminus in *trans* (new Figure 3B).

3) The presented experiments strongly suggest, but do not prove, an autoinhibitory mechanism, as there are no experiments that directly address Epe1 conformation. Therefore, claims regarding such a mechanism should be softened. The title could be changed to "Interaction with histone H3 methylated at lysine 9 regulates the non-enzymatic functions of the putative histone demethylase Epe1" or something similar, and strong claims like "An auto-inhibitory conformation regulates the non-enzymatic properties of Epe1 through its interaction with Swi6" in the Abstract should be amended or removed.

We appreciate the reviewers’ insights and have amended our manuscript by discussing our results as H3K9 methylation dependent changes in protein-protein interactions between Epe1 and Swi6. Consistent with the suggestions provided, we have revised the title and the Abstract of our revised submission.